



# Implementing a sectional scheme for early aerosol growth from new particle formation in the Norwegian Earth System Model v2: comparison to observations and climate impacts

Sara M. Blichner[1], Moa K. Sporre[2], Risto Makkonen[3,4], and Terje K. Berntsen[1]

[1]Department of Geosciences, University of Oslo, Oslo, Norway
[2]Department of Physics, Lund University, Lund, Sweden
[3]Institute for Atmospheric and Earth System Research / Physics, Faculty of Science, University of Helsinki, Finland
[4]Climate System Research, Finnish Meteorological Institute, Helsinki, Finland

**Correspondence:** Sara Marie Blichner (s.m.blichner@geo.uio.no)

**Abstract.** Aerosol-cloud interactions contribute with a large portion of the spread in estimates of climate forcing, climate sensitivity and future projections. An important part of this uncertainty is how much new particle formation (NPF) contributes to cloud condensation nuclei (CCN), and furthermore, how this changes with changes in anthropogenic emissions. Incorporating NPF and early growth in Earth System Models (ESMs) is, however, challenging both due to uncertain parameters (e.g. partici-

pating vapours), structural challenges (numerical description of growth from $\sim$1 to $\sim$100 nm), and due to large scale of ESM grid compared to NPF scale.A common approach in ESMs is to represent the particle size distribution by a certain number of log-normal modes. Sectional schemes on the other hand, where the size distribution is represented by bins, are considered closer to first principles because they do not make an a priori assumption about the size distribution.

In order to improve the representation of early growth, we have implemented a sectional scheme for the smallest particles

(5–39.6 nm diameter) in the Norwegian Earth System Model (NorESM), feeding particles into the original aerosol scheme. This is, to our knowledge, the first time such an approach has been tried. We find that including the sectional scheme for early growth improves the aerosol number concentration in the model when comparing against observations, particularly in the 50–100 nm diameter range. Furthermore, we find that the model with the sectional scheme produces much less particles than the original scheme in polluted regions, while it produces more in remote regions and the free troposphere, indicating a potential

impact on the estimated aerosol forcing. Finally, we analyse the effect on cloud-aerosol interactions and find that the effect of changes in NPF efficiency on clouds is highly heterogeneous in space. While in remote regions, more efficient NPF leads to higher cloud droplet number concentration (CDNC), in polluted regions the opposite is in fact the case.

## 1   Introduction

The formation of new particles in the atmosphere, known as new particle formation (NPF) occurs through the clustering and

nucleation of low volatile vapours. These particles can then influence the climate by growing via condensation to sizes where they act as cloud condensation nuclei (CCN) (Twomey, 1974; Albrecht, 1989) – or even by interacting directly with radiation if they grow large enough (Boucher et al., 2013). NPF has received increasing attention in recent years due to the aforementioned





climate impacts as well as its implications on human health. This has lead to new insights into the mechanisms involved in NPF and subsequently, new parameterization schemes have been developed and included in Earth System Models (ESMs). For example, Gordon et al. (2016) showed that including a NPF pathways from pure organic nucleation nucleation (Kirkby et al., 2016; Riccobono et al., 2014; Gordon et al., 2017, 2016; Dunne et al., 2016; Tröstl et al., 2016) in a global aerosol model, resulted in a considerable diminishing of the estimated negative forcing due to aerosol–cloud interactions since pre-industrial times (+0.22 W/m², 27 %). This result illustrates the importance of adequately representing the effects of NPF in ESMs for our understanding of historical forcing and thus climate sensitivity, especially considering that cloud–aerosol interactions are estimated to be responsible for a large fraction of the observed negative radiative forcing since pre-industrial times (Boucher et al., 2013).

In spite of NPF being subject to a lot of research over the recent years, there is still uncertainty about the species involved in both nucleation, and the subsequent particle growth (Kerminen et al., 2018; Lee et al., 2019). In order for successful NPF, particles must form and grow up to a decent size, often defined to be out of the nucleation mode, i.e., 10 nm. Due to the Kelvin effect, only atmospheric gases with very low volatility are able to contribute to the inital steps on NPF, and in many atmospheric conditions the growth rates provided are too slow for for particles to survive losses to coagulation and evaporation (Semeniuk and Dastoor, 2018). Sulphuric acid is known to be the most important species for nucleation due to its low vapor pressure, while bases such as amines and ammonia may enhance the nucleation rate (Lee et al., 2019; Kerminen et al., 2018). There is evidence that extremely low volatile organic vapors also contribute significantly, especially in remote areas (Semeniuk and Dastoor, 2018; Dunne et al., 2016; Riccobono et al., 2014). For the subsequent growth of the particles, the Kelvin effect decreases and condensing organics of higher volatility, predominantly originating from the oxidation of biogenic volatile organic compounds (BVOCs), become more and more dominant and are essential in most environments (Riipinen et al., 2011; Tröstl et al., 2016).

During all the stages of particle growth, the particles are subject to coagulation, reducing the number of particles that form and that grow to sizes where they can act as CCN ($\sim 50$ nm in diameter (Kerminen et al., 2012)). The majority of this coagulation will occur with particles that are already in the CCN size range, and thus results in a net loss of particles that could eventually act as CCN. However, when two small particles (below the CCN size range) coagulate, this contributes to growth of the combined particle which could then become a CCN (e.g. Kerminen et al. (2018); Lee et al. (2013); Schutgens and Stier (2014)). This effect though, is only significant in highly polluted regions. The survival rate of NPF-particles to CCN sizes is therefore in general depend on a competition between the particle growth rate by condensation and the coagulation sink.

The formation of new particles is tightly constrained by negative feedbacks. If NPF is high, the result will be an increase in particle number and with it, an increase the available surface area for condensation. This will lead to an increase in both condensation– and coagulation sink, which further decreases the growth rate and increases the coagulation sink of new particles forming. The result is then a supression of further NPF (Westervelt et al., 2014, 2013; Semeniuk and Dastoor, 2018; Carslaw et al., 2013; Kerminen et al., 2018; Schutgens and Stier, 2014, etc). These loss processes which constrain the survival of new particles to larger sizes may in fact often be more important than nucleation rate in itself. For example, Carslaw et al. (2013), show that Global Model of Aerosol Processes (GLOMAP) has low sensitivity of number of particle larger than 50 nm to nucleation rate parameterizations, but a high sensitivity to processes affecting the coagulation loss of newly formed particles.



This underlines the importance of adequately representing the processes that constrain the formation of new particles. If not we could end up with models where both aerosol number concentration and CCN are over-sensitive to changes in emissions.

While there is a large body of work on describing when NPF happens in many individual environments, the transferal of this to a generalized context (which is what is needed for a climate model), is very uncertain. In other words, based on knowledge of what drives NPF in a specific environment it is not easy to derive a general parameterization (Kerminen et al., 2018; Lee et al., 2019).

    In the perspective of an ESM, aerosols only become relevant when they approach $\sim 50\,\mathrm{nm}$ in diameter and start to become

relevant as CCN (Kerminen et al., 2012). However, because the formation of particles in this size range is highly dependent on aerosol dynamics at smaller sizes, climate models need to treat these dynamics with a sufficient degree of accuracy. Since climate models are required to run hundreds of years of simulations within a reasonable time span, this involves a trade-off between representing the physical process to the best of our scientific understanding on one hand, and computational cost on the other hand.

In ESMs, it is common to use modal schemes to represent the particle size distribution – i.e. describing the distribution as the sum of some number of log-normal modes (Stier et al., 2005; Liu et al., 2005; Mann et al., 2010; Vignati et al., 2004, etc.). On the other hand, sectional schemes – where the size distribution is represented by bins (Spracklen et al., 2005; Kokkola et al., 2008, etc.) – are in general considered closer to first principles because they do not make an a priori assumption about the size distribution. Nevertheless, modal schemes are generally favored in ESMs because they require fewer tracers and are

much cheaper computationally.

    Any size resolving aerosol scheme must have a cut-off diameter where explicit modelling of aerosol number, growth and losses begin. One natural choice is the size of the critical cluster, around $1\,\mathrm{nm}$ (Lee et al., 2013). While this means that the entire size distribution of particles is treated, it adds disproportionate computational cost to the simulation for aerosols with a very short atmospheric lifetime (both due to growth out of the size range and high sensitivity to coagulation) (see e.g. Lee

et al. (2013)). An alternative is to parameterize the growth and coagulation loss of particles up to a larger diameter, which is the approach used in most ESMs (Kerminen and Kulmala, 2002; Kerminen et al., 2004; Lehtinen et al., 2007; Anttila et al., 2010). These methods involve estimating the flux or the formation of particles at the cut-off diameter, be it modal or sectional, based on estimated growth rate and coagulation sink (see details in methods).

    There are several drawbacks of this approach, especially if the chosen cut-off diameter is high. The most important one is

that it assumes steady state, i.e. the same constant growth rates from the particle is formed up to the cut-off value, which in reality could take several time steps and long enough for conditions to change substantially (hours). A particle may form under conditions with a high growth rate, but in the time it would take for the particle to grow to the cut-off diameter, the growth rate might decrease due to an increased condensation sink by the many new particles being formed. In a model with a relatively high cut-off, this would lead to an overestimation of the growth rate of the nucleated particle, which would in turn lead to

an overestimation of the formation rate at the cut-off (Olenius and Riipinen, 2017; Lee et al., 2013). Olenius and Riipinen (2017) test the effect of the cut-off diameter by explicitly modelling the formation of particles from vapour molecule to $10\,\mathrm{nm}$ diameter and find an over-prediction of a factor of two to orders of magnitude. Similarly, Lee et al. (2013) suggest that during





nucleation events, the smallest particles (<10 nm) can be a significant condensation sink, thus regulating the nucleation via reduced concentrations of precursors. They investigate the effects of cut-off diameter with a sectional aerosol scheme in the

GISS-TOMAS model, and compare 1 nm cut-off with 3 nm and 10 nm cut-offs using Kerminen et al. (2004) to parameterize the survival of nucleated particles to the cut-off. They find that the using a 10 nm cut-off leads to an overestimation of CCN at 0.2 % supersaturation, with 10–20 % overestimation in the surface layer in most of the northern hemisphere, while the globally averaged change to CCN(0.2%) is minor. Furthermore, a 10 nm cut-off produces a high bias in the concentration of particles larger than 10 nm ($N_{10}$) of up to a factor of 3–5 in regions with high nucleation. In addition, they find that the 10 nm cut-off

is sensitive to the time step.

Another drawback of a high cut-off diameter is that most of these parameterizations neglect self coagulation within the sub-cut-off size range, which can be an important growth mechanism during intense new particle formation events. This concern is, however, taken into account in the Anttila et al. (2010) parameterization.

Finally, if the cut-off diameter is high, the time and location where the new particles are inserted into the aerosol model may

be effected since the parameterized growth would add the particles, at the cut-off size, in the same time step as they would be formed, i.e. within $\sim 0.5$ hour. In reality, this growth could take several hours to days, depending on location, at which point the airmass may have moved considerably. This is in particular the case of a high cut-off value, like in NorESM (23.6 nm) (Kirkevåg et al., 2018).

In order to improve the representation of early particle growth, we have implemented a sectional scheme for the smallest particles (5–39.6 nm diameter) in the aerosol scheme in the Norwegian Earth System Model (NorESM). The sectional scheme acts as an intermediate step during NPF and feeds the grown particles into the original modal scheme. This is, to our knowledge, the first time such a hybrid approach has been attempted. The sectional scheme currently involves two condensing species (sulphuric acid and low volatility organics) and 5 bins. The aerosol scheme with these changes will be referred to as

OsloAeroSec. A schematic of the changes from the OsloAero (the original model) to OsloAeroSec is shown in Fig. 1. The motivation is

1. In the original modal scheme in NorESM, the smallest mode has an initial mean radius of 23.6 nm. Particles from new particle formation are inserted into this mode using the parameterization from Lehtinen et al. (2007). It thus does not take into account dynamics within the sub-23.6 nm range (e.g. competition for condensing vapours and growth of particles

over more than one time step).

2. Including a sectional scheme for this size range brings the modelling of early growth closer to first principals while keeping an acceptable computational cost because the number of species involved is low. A sectional scheme within this range represents a good alternative to a nucleation mode which is known to have problems with transferring particles to the larger mode, due to the addition of new particles reducing the median radius of the mode.

In the following we start by describing the aerosol scheme in NorESM (section 2.1) and then the newly implemented sectional scheme for early growth (section 2.2). Next, in section 4.1, we show that the new scheme gives improvements in the





CCN relevant particle number concentration and sizedistribution when compared to observational data from Asmi et al. (2011) consisting of 24 stations in Europe and compiled as part of the EUSAAR project. Finally, we present the global changes in the state of aerosols and following cloud properties in the model with the new scheme (OsloAeroSec) compared to the original model (section 4.2).

## 2   Model description

We start by briefly describing the Norwegian Earth System Model (NorESM) in general before giving a detailed description of its aerosol model, OsloAero, in section 2.1. After this in section 2.2, we will describe what changes to said aerosol scheme that have been introduced in OsloAeroSec. In general, the aerosol scheme after NPF and early growth is left as it is. The only exception to this is that we have also included some changes to the diurnal variability of OH, described in section 2.3.

The Norwegian Earth System Model version 2 (NorESM2) (Seland et al., 2020b; Bentsen et al., 2013; Kirkevåg et al., 2013; Iversen et al., 2013) is largely based on the Community Earth System Model (CESM) version 2 (Danabasoglu et al., 2020; Neale et al., 2012). The aerosol scheme in CESM2 is replaced by OsloAero6 (described below) (Kirkevåg et al., 2018) and the atmospheric component is thus named CAM-Nor6. Furthermore, the ocean model in CESM2 is replaced by Bergen Layered Ocean Model (BLOM) (Seland et al., 2020b), though this is not used in this study as all simulations are run with prescribed sea surface temperature (SST) and sea ice concentrations. The land model is, as in CESM2, is the Community Land Model (CLM) version 5 (Lawrence et al., 2019).

### 2.1   OsloAero: Aerosol scheme in NorESM

The aerosol scheme in NorESM, OsloAero, is a production tagged aerosol model. The most notable difference to other aerosol models is that the aerosol mass is divided into "background" tracers and "process" tracers. The background tracers form log-normal modes which decide the number concentration, while the process tracers alter this initial log normal distribution and their chemical composition. Examples of background tracers are dust, sea salt or particles from NPF, while examples of process tracers are sulphate condensate, sulphate coagulate and organic condensate. After the process tracers are applied, the resulting distribution of the "mixtures" are not (necessarily) log normal anymore. The mass of the tracers is tracked, and the size distributions for cloud activation and optical properties are calculated using a look-up table approach (Kirkevåg et al., 2018).

#### 2.1.1   Chemistry:

CAM-Nor6 has a simplified chemistry scheme for sulfur and organic species, using the chemical pre-processor MOZART (Emmons et al., 2010). Pre-calculated monthly mean oxidant fields consisting of OH, $O_3$, $NO_3$ and $HO_2$ are read from file (for discussion see Karset et al. (2018)).

Condensing tracers in the model are $H_2SO_4$ and two tracers of organics produced by the oxidation of BVOCs, low volatility organics ($SOAG_{LV}$) and semi-volatile organics ($SOAG_{SV}$). The model treats both organic tracers as non-volatile during con-



densation, but represents the volatility by separating which processes each tracer can contribute to: $SOAG_{LV}$ can contribute in new particle formation (NPF) and early growth, while $SOAG_{SV}$ only contributes to condensational growth.

$H_2SO_4$ is emitted directly or produced from oxidation of $SO_2$ by OH or aqueous-phase oxidation by $H_2O_2$ and $O_3$ (Tie et al., 2001). $SO_2$ is either emitted directly or produced by oxidation of DMS. The condensing organic tracers, $SOAG_{LV}$ and $SOAG_{SV}$, are formed from oxidation isoprene and monoterpenes. The emissions of isoprene and monoterpene are calculated online in each time step using the Model of Emissions of Gases and Aerosols from Nature version 2.1 (MEGAN2.1) (Guenther et al., 2012) which is incorporated into CLM5. The atmospheric tracer includes only one tracer for monoterpenes, and thus the

emissions of 21 monoterpene species from MEGAN2.1 are lumped together (Kirkevåg et al., 2018). In addition, production of methansulfonic acid (MSA) by oxidation of DMS is taken into account, but since the model lacks a tracer for MSA, 20% of the MSA is put in the $SOAG_{LV}$ tracer and 80% in the $SOAG_{SV}$.

For complete overview of reactions and reaction rates, see Table 2 in Karset et al. (2018).

### 2.1.2 Condensation:

Following is a description of the condensation routine in chronological order within one time step. The production rate, $P_{gas}$, of a condensing gas is calculated in the gas phase chemistry (section 2.1.1 and the condensation sink, $L_{cond}$ [1/s], is calculated based on the surface area of the background aerosols. Finally, using the initial concentration of the gas, $C_{old}$, from the previous time step, an intermediate concentration, $C_{int}$, is derived by solving the discrete Euler backwards equation,

$$\frac{C_{int} - C_{old}}{\Delta t} = P_{gas} - L_{cond}C_{int} \tag{1}$$

$$C_{int} = \frac{C_{old} + P_{gas}\Delta t}{1 + L_{cond}\Delta t}. \tag{2}$$

This intermediate concentration is then used in the formation of new particles (described in the next section). The NPF subroutine returns an intermediate nucleated mass loss rate, $J_{m,nuc}$. This nucleated mass is then used to calculate a nucleation loss rate, $L_{nuc}$ [1/s]:

$$L_{nuc} = \frac{J_{m,nuc}}{C_{int}} \tag{3}$$

The new gas concentration, $C_{new}$, is calculated by solving the discrete Euler backwards equation again, including the loss rate to nucleation:

$$C_{new} = \frac{C_{old} + P_{gas}\Delta t}{1 + L_{cond}\Delta t + L_{nuc}\Delta t} \tag{4}$$

Finally, the total gas lost to condensation and nucleation, $\Delta C$, is calculated by

$$C_{new} - C_{old} = P_{gas}\Delta t - \Delta C \tag{5}$$

$$\Delta C = P_{gas}\Delta t + C_{old} - C_{new} \tag{6}$$

This condensate/nucleate, $\Delta C$, is then transferred to the corresponding process tracer for condensate of the species (e.g. sulphur condensate) and the background tracer for new particle formation particles. The mass transfer is done based on their





relative contribution to the total loss rate – i.e. the fraction that is moved to the NPF tracer is $f_{\mathrm{nuc}} = L_{\mathrm{nuc}}/(L_{\mathrm{nuc}} + L_{\mathrm{cond}})$ and the fraction to condensation is $f_{\mathrm{cond}} = 1 - f_{\mathrm{nuc}}$.

### 2.1.3 New particle formation:

The tracers contributing to NPF are $H_2SO_4$ and organics (see Makkonen et al. (2014)). As mentioned above, $SOAG_{SV}$ does not contribute to new particles formation. In addition, only half of the $SOAG_{LV}$ concentration in each time step is assumed to be low volatility enough to contribute, and this fraction will be denoted as ELVOC in the following. The nucleation rate is parameterized with Vehkamäki et al. (2002) for binary sulfuric acid-water nucleation in the entire atmosphere and in addition, equation 18 from Paasonen et al. (2010) is added to represent boundary layer nucleation. The Paasonen et al. (2010, eq.18) parameterization is as follows:

$$J_{\mathrm{nuc}} = A_1[H_2SO_4] + A_2[\mathrm{ELVOC}] \tag{7}$$

where $J_{\mathrm{nuc}}$ [1/s] is the nucleation rate and $A_1 = 6.1 \times 10^{-7}$ s$^{-1}$ and $A_2 = 3.9 \times 10^{-8}$ s$^{-1}$.

The survival of particles from nucleation at $d_{\mathrm{nuc}} \approx 2$ nm, to the background mode holding the NPF particles, number median diameter 23.6 nm, is parameterized by Lehtinen et al. (2007). The formation rate, $J_{d_{\mathrm{mode}}}$ of particles at the smallest mode is calculated by

$$J_{d_{\mathrm{mode}}} = J_{\mathrm{nuc}} \exp\left(-\gamma d_{\mathrm{nuc}} \frac{CoagS(d_{\mathrm{nuc}})}{GR}\right) \tag{8}$$

where, $d_{\mathrm{nuc}}$ is the diameter of the nucleated particle, $CoagS(d_{\mathrm{nuc}})$ is the coagulation sink of the particles [h$^{-1}$], $GR$ is the growth rate [nm/h] of the particle (from $H_2SO_4$ and ELVOC, calculated using eq. 21 from Kerminen and Kulmala (2002)) and $\gamma$ is a function of $d_{\mathrm{mode}}$ and $d_{\mathrm{nuc}}$:

$$\gamma = \frac{1}{m+1}\left[\left(\frac{d_{\mathrm{mode}}}{d_{\mathrm{nuc}}}\right)^{(m+1)} - 1\right], \; m = -1.6. \tag{9}$$

Furthermore, $CoagS(d_{\mathrm{nuc}})$ is calculated from $CoagS(d_{\mathrm{mode}})$ assuming a power-law dependency on diameter, $CoagS(d_{\mathrm{nuc}}) = CoagS(d_{\mathrm{mode}}) \cdot \left(\frac{d_{\mathrm{nuc}}}{d_{\mathrm{mode}}}\right)^m$ (Lehtinen et al., 2007, eq. 5).

Since Kirkevåg et al. (2018), we have developed an improvement to the new particle formation rate (also used in Sporre et al. (2019, 2020)). The $CoagS(d_{\mathrm{nuc}})$ previously included only coagulation onto accumulation and coarse mode particles, but we amended this to include coagulation onto all pre-existing particles. This modification gives a lower and more realistic survival rate of particles from formation at 2 to 23.6 nm.

### 2.1.4 Coagulation:

OsloAero takes into account coagulation between Aitken mode particles and accumulation- and coarse mode particles, with coagulation coefficients from the Fuchs form for Brownian diffusion (section 12.3 in Seinfeld and Pandis (1998)). Technically, a normalized coagulation sink is calculated for each relevant combination of background modes, assuming some fixed prior





condensation/coagulation growth. To compute the normalized coagulation sink, the size distribution is split into 44 bins for the coagulation receiver mode (the larger particle) and a coagulation sink with each bin is calculated and normalized by the number concentration. This way, the normalized coagulation sink only has to be computed once. In addition, coagulation of aerosols with cloud droplets is estimated. See Seland et al. (2008) for more detail.

## 2.2 OsloAeroSec: New sectional scheme

The purpose of introducing the sectional scheme is to get a more realistic growth and loss dynamic within the smallest aerosol sizes, with the aim of better modelling aerosol–climate effects. These smallest particles have insignificant effects on climate directly, but rather play a role through how they affect the size distribution of the larger particles. For this reason, we do not let the aerosols in the sectional scheme directly affect the radiation and cloud parameterizations, but rather consider only how new particle formation through nucleation, condensation and coagulation affect the larger aerosols in the modal scheme.

The sectional scheme currently consists of five bins (though this is flexible) and the bin sizes are set according to a discrete geometric distribution – the volume-ratio distribution (Jacobson, 2005, sec.13.3) – as follows: Let $d_1, d_2, \ldots, d_5$ be the diameter for each bin and $v_1, v_2, \ldots, v_5$ be the volume per particle for each bin. Each particle in the bin is assumed to have this same volume (Jacobson, 2005). The volume-ratio distribution ensures that the volume per particle ratio between adjacent bins is fixed, i.e.,

$$r_v = \frac{v_{i+1}}{v_i} \tag{10}$$

is fixed. This gives that the ratio between the diameter in adjacent bins, $r_d$ will be:

$$r_d = \frac{d_{i+1}}{d_i} = (r_v)^{1/3}. \tag{11}$$

Particles are moved into the original aerosol scheme in the NPF background mode when they reach $d_{max} = 39.6$ nm which is the volume median diameter of this mode. The volume median diameter is chosen to preserve both number and mass of the particles. Note that $d_{max}$ is the diameter where the particles are moved to the modal scheme. The choice of $d_{min}$, the smallest diameter bin, is flexible, and we have chosen 5 nm here. So for number of bins, $N$,

$$r_d = \left(\frac{d_{max}}{d_{min}}\right)^{\frac{1}{N}}, \tag{12}$$

where $d_{max} = 39.6$ nm, $d_{min} = 5$ nm and $N = 5$.

The sectional scheme includes condensation from two precursors, $H_2SO_4$ and $SOAG_{LV}$, while $SOAG_{SV}$ is considered not low volatile enough. This gives a total of $N$ (number of bins) $\times 2$ tracers for the model to keep track of, keeping computational costs reasonable.

### 2.2.1 Nucleation:

Nucleation is still parameterized with Vehkamäki et al. (2002) for binary sulfuric acid-water nucleation in the entire atmosphere, the boundary layer nucleation has been updated from (Paasonen et al., 2010, eq.18)(see eq. 7) to Riccobono et al.





(2014):

$$J_{\text{nuc}} = A_3 [\text{H}_2\text{SO}_4]^2 [\text{ELVOC}]$$ (13)

where $A_3 = 3.27 \times 10^{-21}$ cm$^6$ s$^{-1}$

The update was done both due to the Riccobono et al. (2014) parameterization being based on later and thus more recent research and due to the fact that NPF was too high and lasting too long with the Paasonen et al. (2010) parameterization in CAM6-Oslo. The rate at which particles are introduced into the smallest bin, $J_{d_{\min}}$, is still parameterized with eq. 8 defined above (Lehtinen et al., 2007), but with $d_{\text{form}} = d_{\min}$ so that the cut-off size smaller than before.

### 2.2.2   Condensation

The condensation is done in the same way as for OsloAero6, except that the calculated loss rate to condensation $L_{\text{cond}}$ now is the sum of loss to condensation onto the background modes from OsloAero and the condensation onto the sectional bins, $L_{\text{cond}} = L_{\text{cond,modes}} + L_{\text{cond,sec}}$, in equations 2 and 4. Furthermore, the total gas lost, $\Delta C$, calculated by eq. 6, is then distributed as

$$f_{\text{nuc}} = \frac{L_{\text{nuc}}}{L_{\text{nuc}} + L_{\text{cond,modes}} + L_{\text{cond,sec}}}$$ (14)

$$f_{\text{cond,sec}} = \frac{L_{\text{cond,sec}}}{L_{\text{nuc}} + L_{\text{cond,modes}} + L_{\text{cond,sec}}}$$ (15)

$$f_{\text{cond,modes}} = \frac{L_{\text{cond,modes}}}{L_{\text{nuc}} + L_{\text{cond,modes}} + L_{\text{cond,sec}}}$$ (16)

where $f_{\text{nuc}} + f_{\text{cond,sec}} + f_{\text{cond,modes}} = 1$. In other words, the condensate added to the modes is $C_{\text{lost,tot}} \cdot f_{\text{cond,modes}}$. In the same fashion, condensing mass to the sectional scheme is distributed to the different bins by the strength of their respective conden-

sational sinks:

$$f_{bin(d_i)} = f_{\text{cond,sec}} \cdot \frac{L_{\text{cond,bin}(d_i)}}{L_{\text{cond,sec}}}$$ (17)

so that the condensate added to any bin, $d_i$, is equal to $\Delta C \cdot f_{\text{cond,bin}(d_i)}$.

Finally, the condensational growth of particles within the sectional scheme is done in quasi-stationary structure (Jacobson,

1997), meaning the particles grow in volume but are fitted back onto the full stationary grid between each time-step (Jacobson, 2005, sec 13.3). This is done by assuming that (1) the total volume is constant before and after the transfer between the bins, and (2) the total number is the same. Let $v_i$ and $v_{i+1}$ be the volume of a particle in bin $i$ and the next bin, $i+1$, priory to any growth. Let $v_i'$ be the volume of a particle in bin $i$ after growth. Furthermore, let $N_i$ be the number of particles in bin $i$ priory to growth and $\Delta N_{i+1}$ be the number of particles moved to the next bin $i+1$. Since we do not have any evaporating species,

we can easily solve the equation conserving both number and volume of aerosol for each species:

$$v_i' N_i = v_i (N_i - \Delta N_{i+1}) + v_{i+1} \Delta N_{i+1}$$ (18)





and solving for $\Delta N_{i+1}$ gives

$$\Delta N_{i+1} = N_i \cdot \frac{v_i' - v_i}{v_{i+1} - v_i}. \tag{19}$$

After the particle mass is moved in this way, the freshly nucleated particles from the same time step are added to the smallest

bin. The rationale behind this is that the nucleated particles in the same time step do not take part in the condensation sink calculation, and thus including them before the redistribution of mass on the sectional grid, would only imply adding particles with no added condensate.

### 2.2.3 Coagulation:

In addition to the unchanged coagulation in the original OsloAero scheme (see section 2.1.4), we calculate the coagulation

sink of the sectional particles onto all larger particles. This is done in the same way between particles in the original OsloAero scheme, in that a normalized coagulation sink is calculated for each background mode, by dividing the size distribution into 44 bins. When sectional particles coagulate with particles in the "modal" scheme, their mass is transferred to the corresponding process tracer for condensate. This is done for simplicity and because the alternative would be to place them in the coagulation tracers – one of the process tracers – in the original scheme, which will only contribute to changes in the larger particles.

In addition to this, coagulation between the particles in the sectional scheme is taken into account. When two particles in the sectional scheme collide, this results in the loss of the particle in the smaller bin, and the addition of mass to the particle in the larger bin. After this is done in each time step, the mass in the sectional scheme is redistributed in the same way as after condensation (see previous section).

### 2.3 Chemistry: changes to oxidant diurnal variation:

The oxidant concentrations of hydroxyl radical (OH), nitrate radical (NO$_3$), hydroperoxy radical (HO$_2$) and ozone (O$_3$) in the model are prescribed by 3D monthly mean fields (see Seland et al. (2020b)). On top of this, a diurnal cycle is applied to OH, HO$_2$ and NO$_3$. In the default version of the model, the diurnal cycle for OH basically a step function based on whether it is before or after sunrise. Since OH in particular is very important for the diurnal cycle of H$_2$SO$_4$, this leads to more or less a step function in H$_2$SO$_4$ concentrations as well, which is not very realistic in terms of NPF. We therefore implemented a simple

sine shape to the daily variation in place of the step function.

### 3 Model simulations and output post processing

### 3.1 Simulation description

In the following analysis we include simulations with three versions of the CAM6-Nor.

– A simulation with OsloAeroSec, referred to simply as "OsloAeroSec" (see sec. 2.2)

– A simulation with the default version of OsloAero (see sec.2.1), referred to as "OsloAero$_{def}$"





– A simulation with the default version of OsloAero, but with the same changes to the nucleation rate (eq. 13) and oxidants (see sec. 2.3) as OsloAeroSec. Referred to as "OsloAero$_{imp}$".

The last simulation, OsloAero$_{imp}$, is added in order to separate the changes done in OsloAeroSec to the nucleation rate and the diurnal concentration in the oxidants (described above) to the effect of adding a sectional scheme. The simulation characteristics
are also summarized in Table 1.

NorESM2 is run with CAM6-Nor (release-noresm2.0.1, https://github.com/NorESMhub/NorESM) Kirkevåg et al. (2018) coupled to the Community Land Model version 5 (CLM5) (Lawrence et al., 2019) in BGC (biogeochemistry) mode and prognostic crops. We use prescribed sea surface temperature (SST) and sea ice concentrations at $1.9 \times 2.5$ ° resolution (Hurrell et al., 2008). Simulations are run from 2007 and throughout 2014 with CMIP6 historical emissions and greenhouse gas con-
centrations (Seland et al., 2020b) and nudged meteorology (horizontal wind and surface pressure) to ERA-Interim (Berrisford et al., 2011) using a relaxation time of 6 hours (Kooperman et al., 2012) (as described in Karset (2020, sec 4.1)). The year 2007 is discarded as spin-up. The initial conditions for all simulations are taken from a simulation with CAM6-Nor run from 2000 and throughout 2006.

### 3.2 Post-processing of model output

All figures, except comparisons to observations (described below), are produced from monthly mean output files from the model. When we present figures showing averaged values over maps, these are either column burdens or "near-surface" averages of the variable in question. The "near-surface" averages are calculated as the average of all grid cells below 850 hPa, weighted by the grid cell pressure thickness to account for the mass in the grid cell. Cloud radiative effects and direct radiative effects are calculated as described in (Ghan, 2013).

For the model to model comparisons, we include an analysis of whether the change is significant. Dots are included in the plots to indicate where the difference between the two models is significant with a two-tailed paired Student's t–test with 95 % confidence interval.

When we compare the model runs, we compare model version with and without an explicit treatment of the smallest particles. We therefore introduce the following subgroups of particle number concentration. We refer to particle number concentrations
excluding particles in the sectional scheme, as $N_a$. This includes all the particles for the OsloAero simulations (OsloAero$_{def}$ and OsloAero$_{imp}$), but excludes the particles still in the sectional scheme for OsloAeroSec. Furthermore, the total number of aerosols, we refer to as $N_{tot}$, and the concentration of aerosols in the sectional scheme will be referred to as $N_{sec}$. Finally, the aerosol scheme also tracks the number of particles in the modal scheme originating from NPF, and this we denote by $N_{NPF}$. This is summed up in Table 3. Note that changes in $N_{NPF}$ and $N_a$ in general follow the same patterns, because we do not
introduce changes to other particles than those from NPF.

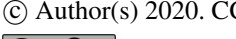



### 3.3 Processing of model output data prior to comparison with observations

We compare the nudged model simulations for years 2008 and 2009 to observed size distributions from the EUSAAR dataset from Asmi et al. (2011). The dataset contains time series of hourly data for number concentrations of particles with diameter between 30 and 50 nm ($N_{30-50}$), 50 and 500 nm ($N_{50-500}$), 100 and 500 nm ($N_{100-500}$) and finally 250 and

500 nm ($N_{250-500}$). In this study, we focus on the concentration of particles with diameter between 50 and 100 nm, i.e. $N_{50-100} = N_{50-500} - N_{100-500}$. Throughout the simulation period, we output hourly mean values describing the modelled size distribution.

The model outputs a log-normal fitting to the size distribution in terms of parameters for 12 log-normal modes. In other words, the total size distribution is

$$\frac{dN}{d(d_p)} = \sum_{i}^{12} \frac{dN_i}{d(d_p)}.$$
(20)

Each term $\frac{dN_i}{d(d_p)}$ is furthermore defined in terms of output parameters from the modal number median diameter, $d_{m,i}$, geometric standard deviation, $S_i$, and the number concentration in the mode, $N_i$:

$$\frac{dN_i}{d(d_p)} = \frac{N_i}{d_p log(S_i)\sqrt{2\pi}} exp\Big(-\frac{(log(d_p) - log(d_{m,i}))^2}{2 log(S_i)}\Big).$$
(21)

For each mode, we can then calculate the number of particles in a size from diameter $d_1$ to $d_2$ by

$N_{i,d_1-d_2} = N_i(d < d_2) - N_i(d < d_1)$
(22)

where $N_i$ is the cumulative distribution function of the distribution in eq. 21, thus

$$N_i(d < x) = \frac{1}{2} + \frac{1}{2}\text{erf}\Big[\frac{log(x) - log(d_{m,i})}{\sqrt{2} log(S_i)}\Big].$$
(23)

The total number concentration in a size range is thus $N_{d_1-d_2} = \sum_{i=1}^{12} N_{i,d_1-d_2}$. We calculate these variables for each hour and do further statistics on the result. By using such a fine time resolution, we avoid a common imprecision arising when

averaging the parameters of the size distribution, $r_{m,i}$ and $S_i$, over a longer time period (i.e. monthly output).

Furthermore, for the comparison of size distributions, we calculate $\frac{dN}{dlog(d_p)} = d_p\frac{dN}{d(d_p)}$ for an array of diameters and do further statistics on the hourly values.

## 4 Results and discussion

### 4.1 Comparison to EUSAAR dataset

In this comparison we focus on $N_{50-100}$ because particles smaller than 50 nm are unlikely to be relevant for CCN and particles above 100 nm are less effected by the changes to the NPF scheme (see e.g. the size distributions in Fig. 6).





Figures 2 and 3 show the distribution of the modelled minus the observed values for $N_{50-100}$ in hourly resolution and with all valid station data included.

From Fig. 2 we can see a clear improvement with OsloAeroSec, compared to both OsloAero$_{def}$ and OsloAero$_{imp}$. The improvement is most pronounced in summer, where OsloAero$_{def}$ and OsloAero$_{imp}$ overestimate $N_{50-100}$, while it is also clear in autumn and spring. It is also encouraging that OsloAeroSec has a clear decrease in the times when the number concentration is highly over-estimated, while there is not a similar increase in times when it is under-estimated. Furthermore, we see that changes to nucleation parameterization and diurnal variation in oxidants in OsloAero$_{imp}$ reduce the bias compared

to OsloAero$_{def}$. In winter, NPF is low, so we see little difference between the different schemes. Figure 3 shows the same as Fig. 2 but for each individual station. OsloAeroSec (OsloAeroSec) shows improvement against OsloAero (OsloAero$_{def}$ and OsloAero$_{imp}$) in most stations during JJA, while sometimes underestimating $N_{50-100}$ in MAM (e.g. VHL, MPZ, HWL).

The annual variability of both models and observations are shown in Fig. 4, where the monthly median (solid line) and percentiles (16th to 84th) are plotted for each station. Again it is clear that OsloAeroSec in general reduces the high bias of

OsloAero$_{def}$ and OsloAero$_{imp}$, and especially when the bias is very high (e.g. OBK, HPB, FKL, ZSF, CMN, BEO). The exceptions that stand out are e.g. CBW, JRC, ZEP and KPO, where all versions of the model do rather poorly, both in absolute numbers and in terms of representing the annual variability. This might indicate that aerosol or precursor emission in the model are not accurate, e.g. due to local sources that are unaccounted for in the model. For CBW, NPF should not be an important source of aerosols during winter and autumn (Mamali et al., 2018), so it is likely that other aerosols are responsible

for the underestimation during these seasons. Dall'Osto et al. (2018) note a strong influence of local anthropogenic emissions at this station, which is likely not captured in the CMIP6 emissions. However, during summer, the model may well show an underestimation of production of particles from NPF which becomes slightly worse with OsloAeroSec. According to Dall'Osto et al. (2018), NPF should be most frequent in JRC and KPO during spring, which the $N_{50-100}$ does not really reflect, probably due to other particles dominating the annual variability. Furthermore, at ZEP station, the concentrations are underestimated in

all months except late autumn and winter. At this station the concentrations in the sectional scheme (see Fig. 6,S7-S10), reveal that there is relatively many particles forming at this location, but they do not survive to 50 nm. All models perform badly here, with OsloAeroSec and OsloAero$_{imp}$ performing slightly worse than OsloAero$_{def}$. In PLA and WAL, the OsloAeroSec results in too low values, while OsloAero$_{def}$ and OsloAero$_{imp}$ perform better. In station MHD, FKL, ZSF, CMN and BEO, the models overestimation of $N_{50-100}$ is reduced OsloAeroSec, but is still significantly too high.

The normalized root mean square error (NRMSE) is improved with OsloAeroSec for both $N_{50-100}$ and $N_{50-500}$, while it stays more or less the same for $N_{100-500}$. The NRMSE is shown in Fig. 5 and is calculated for each season and each model version, using hourly resolution and all available data. The greatest improvement is seen in $N_{50-100}$ and in summer, followed by SON and MAM, while DJF is mostly unchanged. $N_{50-500}$ shows improvement in the same seasons, while there is little change in prediction skill for $N_{100-500}$.

This is likely due to the fact that the particles in CAM6-Nor, are not transferred to larger modes by condensational growth, so in these simulations, changes in $N_{100-500}$ are decided by the tail one NPF-particle mode. Changes are thus determined by





number of particles and how much condensate is added to the mode, i.e. the change in the number median diameter of the mode.

Even though the $N_{50-100}$ improves, Fig. 6 reveals that at the concentrations at smaller sizes are overestimated in most locations. The figure shows the size distribution of particles at each station from both observations and the three versions of CAM6-Nor. For the sectional scheme, the distribution is the sum of particles in the sectional scheme and the modal scheme. This is why it has "spikes", and why there is often a large reduction in $dN/dlog_{10}D$ at the intersection between the sectional scheme and the modal scheme which might be misunderstood to mean that disproportionately many particles are lost in the transition between sectional and modal scheme. The distribution in the sectional scheme, without adding the modal particles, is shown with the dashed line. One important reason why the sectional scheme overestimated the number of particles for this may be that the number of particles above $\sim 100$ nm is underestimated in all the models versions in most of the stations (see e.g. the surface distribution in Fig. S6).

This is particularly pronounced in summer, where the number of particles in the sectional scheme is particularly high (see Fig. S8). Since NPF mostly influences nucleation and Aitken mode particles, this is likely due to other aerosol sources not being adequately represented in the model. This leads to an underestimation of coagulation sink and hence an overestimation of the formation rate. To the same effect, the condensation sink may be too low, again leading to too many new particles forming. This is particularly clear in the arctic station Zeppelin (ZEP), where the measurements show a peak in particles between 100 nm and 200 nm, which are completely missing in the models. The combination of a too high formation rate, and a slow condensation growth rate, leads to too many particles in the smaller sizes.

Overall, adding the explicit treatment of the smaller particles in OsloAeroSec does improve the representation of CCN relevant particles in the model. We especially reduce number concentrations where they are very highly overestimated.

### 4.2 Comparison to original model:

The following section will present general differences in OsloAeroSec compared to the two versions of the original model, OsloAero$_{def}$ and OsloAero$_{imp}$. For this analysis, we make use of the full global model output in monthly mean resolution. We will start with comparing the particle number concentrations and properties of the aerosols. The original version of the CAM6-Nor aerosol scheme does not explicitly model the smallest particles, so in order to get an apples-to-apples comparison, we focus on properties relevant for climate, as represented by the modal aerosol scheme when comparing OsloAeroSec to OsloAero$_{def}$ and OsloAero$_{imp}$. See table 3 for a summary of the definitions of the variables defining number concentration. We then proceed to changes in cloud properties and finally the radiative effect.

### 4.2.1 Aerosols:

The total number of particles, $N_{tot}$, increases in OsloAeroSec compared to OsloAero$_{def}$ and OsloAero$_{imp}$ due to the addition of particles not explicitly treated before. In Fig. 7 the absolute number of sectional particles, $N_{sec}$, in OsloAeroSec is shown (a and c) together with the total number of particles, $N_{tot}$, (right, b and d). The maps in Fig. 7a and b show near surface averages,





as defined above in section 3.2. As can be seen from Fig.7d, the change is particularly strong in the upper troposphere, where

$N_{tot}$ is very low in OsloAero$_{imp}$ and OsloAero$_{def}$.

Figure 8a shows averaged profiles of $N_a$ for each model version, while b and c show maps of the near-surface relative difference in OsloAeroSec compared to OsloAero$_{def}$ and OsloAero$_{imp}$, respectively. On average, the global near-surface $N_a$ decreases in OsloAeroSec by 15 % compared to OsloAero$_{imp}$ and 36.2 % compared to OsloAero$_{def}$. However, at high latitudes the change relative to OsloAero$_{imp}$ is small, or positive, especially over the southern ocean. When considering the vertical

change shown in Fig. 8 a), OsloAeroSec has less particles close to the surface, while the difference is reduced further up in the atmosphere. In the free troposphere the difference becomes positive, meaning further away from the surface, OsloAeroSec lets more particles survive through early growth. For the global average this happens roughly at 700 hPa, while over ocean, it happens already at 800 hPa. Over the continents, OsloAero$_{imp}$ is always higher, though the difference decreases with height. From these results, we can conclude that on average the sectional scheme produces more particles in more remote regions,

both horizontally and vertically.

In all model versions, the growth of the particles from nucleation to the smallest mode happens by condensation of the two tracers $H_2SO_4$ and $SOAG_{LV}$. The relative contribution of $H_2SO_4$ and $SOAG_{LV}$ to this growth changes with OsloAeroSec, but interestingly, also between OsloAero$_{def}$ and OsloAero$_{imp}$. Figure 9a shows the SOA fraction of the particles that have survived to the modal scheme averaged over regions. Firstly, the SOA fraction goes down in OsloAero$_{imp}$ compared to OsloAero$_{def}$ and

secondly, globally it goes up with OsloAeroSec. We start with exploring the difference between OsloAero$_{def}$ and OsloAero$_{imp}$. These two simulations have the same parameterization for survival of particles from nucleation up to the model scheme (see section 2.2), but OsloAero$_{imp}$ has an improved diurnal variation in the oxidants resulting in a higher, diurnal peak in $H_2SO_4$ (not shown). Additionally, the nucleation parameterization in OsloAero$_{imp}$ is on the form $H_2SO_4$ $^2\times$ELVOC, meaning that as $H_2SO_4$ increases, the nucleation rate increases to the power of two, while in OsloAero$_{def}$ the increase is linear with both

$H_2SO_4$ and ELVOC. Furthermore, because the growth from nucleation to modal scheme happens within one time step in these simulations, the fraction of the growth from SOA is entirely based on $H_2SO_4$ and ELVOC at the moment of nucleation. This means that if most of the particles form when $H_2SO_4$ is at it's highest, $H_2SO_4$ will also dominate the post-nucleation growth. This explains the reduced contribution of SOA in OsloAero$_{imp}$ relative to OsloAero$_{def}$.

The change seen in OsloAeroSec compared to OsloAero$_{def}$ and OsloAero$_{imp}$, on the other hand, can be explained by

two factors: (1) though OsloAeroSec has the same changes to oxidants and nucleation parameterization as OsloAero$_{imp}$, the particles grow in the sectional scheme over more than one time step, and thus be exposed to different concentrations of $H_2SO_4$ and $SOAG_{LV}$. Thus, the concentrations at the time of nucleation will be less dominant for the growth. (2) In OsloAero$_{def}$ and OsloAero$_{imp}$ only ELVOC, which is 50% of the $SOAG_{LV}$, will contribute to growing the particles up to the modal scheme, while in OsloAeroSec 100% of the $SOAG_{LV}$ can contribute after the particles have reached the sectional scheme (5 nm), thus

increasing the SOA fraction. The result is a combination of these effects, in some regions, like over the Amazon, the effect seems to be dominated by the change in nucleation timing such that the SOA fraction goes down compared to OsloAero$_{def}$. In most regions the effect is that the SOA fraction increases.





Note that the changes in hygroscopicity from this are minor and are mitigated by the fact that additional condensate is added to the particles after they reach the modal scheme.

The strength and sign of the change in number concentration between OsloAeroSec and the original model varies with location.

To investigate what conditions lead to the changes in NPF particles, we focus on the difference in $N_{NPF}$ between OsloAeroSec and OsloAero$_{imp}$ and analyse its relationship to relevant variables in OsloAero$_{imp}$. Thus, we can analyse under which conditions in the model (polluted, clean, high NPF etc.) $N_{NPF}$ increases or decreases with the sectional scheme. Figure 10 shows
the relationship for nucleation rate ($J_{nuc}$, a), growth rate (GR, b), $H_2SO_4$ (c), $SOAG_{LV}$ (d), $N_{NPF}$ (e) and coagulation sink for newly formed particles (CoagS, f). This 2D histogram includes each grid cell below 100 hPa and monthly mean values are used for each grid-cell.

Firstly, most of the variables show a branch with a strong negative relationship with the change in $N_{NPF}$ ($\Delta N_{NPF}$). Further investigation shows that the grid-cells that constitute this branch are mainly close to the surface and, as can be seen from Fig. 10e,
where $N_{NPF}$ and CoagS are high. In other words, what we are seeing is that in regions with high CoagS and $N_{NPF}$, the sectional scheme reduces the number of particles that survives drastically and more the higher they were initially in OsloAero$_{imp}$. This resembles what we saw when comparing to station data, where in particular the very high over-estimations were reduced.

For the other grid-cells, where $N_{NPF}$ and CoagS are lower, there is another branch showing a positive relationship with GR, $H_2SO_4$ and $SOAG_{LV}$. From panel e and f, it is clear that these grid-cells have $N_{NPF}$ concentrations under roughly 100 cm$^{-3}$
and CoagS under roughly $10^{-3}$ h$^{-1}$. In this regime the sectional scheme allows more particles to survive, and condensational growth is more important.

In sum this means that in regions with very high number concentrations initially, the sectional scheme reduces the number of particles that survive proportional to the coagulation sink/initial number of particles, while when the number of particles is initially small, the sectional scheme lets more particles survive and the change is more proportional to the concentration of
condensing vapors.

As mentioned before, the Lehtinen et al. (2007) parameterization assumes steady-state GR and CoagS throughout the growth up to the aerosol model cut-off diameter, while in reality the aerosol often forms e.g. when the GR is high and the CoagS is relatively low. The steady-state assumption is likely to give especially biased results in areas with high variability in aerosol and precursor concentration. Since this is especially the case in areas with high aerosol concentration, like the boundary layer,
this may be why it is especially here that the sectional scheme reduces $N_{NPF}$. In the sectional scheme, the particles may grow over some time and space before reaching the modal scheme, and thus experience other concentrations.

### 4.2.2 Cloud–aerosol interactions

The sectional scheme affects the CCN concentrations by influencing the number of particles that survive to the modal scheme, and thus also influences the cloud droplet activation scheme. The changes to cloud properties are shown in Fig. 11 e–h and
Fig. 12. We include variables that indicate changes to cloud properties from cloud aerosol interaction. Unfortunately, CCN calculations are not currently available for CAM6-Nor.





We start by discussing the changes in OsloAeroSec compared to OsloAero$_{imp}$ shown in the right column of Fig. 11 and 12. Figure 11f and h show the change between OsloAeroSec and OsloAero$_{imp}$ in cloud droplet number concentration (CDNC) and $r_e$ averaged over longitude and time. These plots reveal that the CDNC increases and $r_e$ decreases at most latitudes and

heights, except above $\sim 40\,°$N.

Considering the change in N$_{NPF}$ shown in Fig 11b, the change in cloud properties reveals a highly non-linear response in Fig. 11f, where the CDNC increases (and similarly $r_e$ decreases) both where there are more N$_{NPF}$ (high in the southern hemisphere atmosphere) and where there are fewer (near surface in the tropics). To investigate this, we show in Fig. 13a and b the Pearson correlation coefficient between $\Delta$N$_{NPF}$ and $\Delta$CDNC calculated for each latitude and pressure level, along

time (monthly mean) and longitude. The pattern shows that in remote regions, i.e. polar and high troposphere, higher N$_{NPF}$ is positively correlated with higher CDNC, while in less remote regions, the opposite is the case. The correlations are very similar when comparing to OsloAero$_{def}$ (Fig. 13a) and OsloAero$_{imp}$ (Fig. 13b). These regions correspond roughly to regions of low particle concentrations (upper atmosphere) and high particle concentrations (surface). The reason for these correlations is likely that when the number of particles decrease, the amount of condensate available for each particle increases, thus

increasing the number median diameter of each mode. This is seen in Fig. 11b and d, where we have inverse patterns in the difference in N$_{NPF}$ and number median radius for NPF particles (NMR$_{NPF}$). Since decreasing the number of particles in general causes the remaining particles to be bigger, there may be fewer particles in total, but a larger fraction of the ones that are left is likely to activate. This is true in general, but in polluted regions, where there are many particles to compete for the same water vapour, the maximum supersaturation will be lower and the minimum activation diameter will be higher than in remote

regions. Therefore, the change in size of the mode may be more important than the change in number. In more remote regions, the maximum supersaturation will be higher and the activation diameter smaller and thus the number of particles to activate will be more dependent on absolute number in the mode than the number median radius of the mode. Additionally, in highly polluted areas, an increased number of particles may inhibit activation because more particles compete for the same water vapor.

Keeping this in mind, the cloud property changes are easier to explain. When the number concentration decreases in remote regions, the CDNC ($r_e$) increases (decreases) and the opposite for non-remote regions.

In general these results are reflected in Fig. 12 showing the changes in cloud properties on maps. There are significant differences over large parts of the high latitude regions and the Amazon: an increase in column integrated cloud droplet number (col$_{droplets}$, b), a decrease in cloud top effective droplet radius ($r_e$(CT), d) and an increase in total cloud water path (CWP, f).

Note that, there is a reverse pattern over the northern hemisphere continent, where col$_{droplets}$ decreases, $r_e$(CT) increases and CWP decreases.

The difference in N$_{NPF}$ between OsloAeroSec and OsloAero$_{def}$ in Fig. 11a, shows a stronger and more prevalent decrease than the difference between OsloAeroSec and OsloAero$_{imp}$ in b, due to the fact that OsloAero$_{def}$ has, in general, more particles than OsloAero$_{imp}$.





The cloud effects follow closely the same rationale as for OsloAeroSec versus OsloAero$_{imp}$, explained above: the decrease in polluted regions (tropics, close to the surface) give an increase in CDNC, while a decrease in remote regions (northern hemisphere, free troposphere) gives a decrease in CDNC.

    The right column in Fig. 12 shows maps for the relative difference between OsloAeroSec and OsloAero$_{def}$. In this case the hemispheric asymmetry is clearer than for OsloAeroSec versus OsloAero$_{imp}$: in the northern hemisphere above $\sim 30°$, we

have a decrease in col$_{droplets}$ (a), $r_e$ clearly increases (c), CWP decreases (e) and the net cloud effect is a slight warming (g). Over the south pole and large parts of the tropics, the opposite is the case.

    The result is that the cloud effects of particles from NPF may depend highly on where these are formed.

### 4.2.3   Radiative effects

The changes in cloud properties discussed in the section above entail changes to the net cloud radiative effect (NCRE), shown in

Fig. 12g and d. The globally averaged NCRE becomes more negative with OsloAeroSec compared to both OsloAero$_{def}$ ($-0.05$ Wm$^{-2}$) and OsloAero$_{imp}$ ($-0.11$ Wm$^{-2}$). The globally averaged $\Delta$NCRE is less negative for OsloAeroSec–OsloAero$_{def}$, because there are quite strong compensating positive values in the northern mid– to high latitudes.

    Aerosols can scatter or absorb radiation directly and this effect is referred to as the direct aerosol effect. The changes in aerosol size distribution induced by using OsloAeroSec can not only affect the climate through changes in the cloud radiative

effect, but also to a lesser extent through changes in the direct aerosol effect. We calculate the direct aerosol effect by the method of Ghan (2013). The change in direct aerosol radiative effect (DRE) is shown in Fig. 14. In general the change is small with up to $\pm \sim 0.4$ Wm$^{-2}$ regionally and $0.03$ Wm$^{-2}$ and $0.02$ Wm$^{-2}$ globally compared to OsloAero$_{def}$ and OsloAero$_{imp}$, respectively. This is because the influence of the sectional scheme on the particles large enough to interact directly with radiation is rather small. What we do see is likely due to the fact that when number concentrations decrease (increase), we get

an increase (decrease) in condensate available for each particle. Thus more (less) particles grow into the range where they can interact directly with radiation. This is illustrated by the top two rows in Fig. 11 showing N$_{NPF}$ and number median radius of the NPF particles which have inverse patterns and was also seen in (Sporre et al., 2020).

### 5   Implications and further discussion

From the results above, it is clear that including explicit treatment of the early growth in OsloAeroSec does increase prediction

skill compared to the original parameterization for particles above 50 nm in diameter. The difference is largest in summer, where the sectional scheme reduces the number of particles in $N_{50-100}$ substantially, bringing it closer to the observed values. While the overestimation of particles above 50 nm is vastly reduced with OsloAeroSec, there is still a considerable overestimation of the smallest particles (below $\sim 20$nm). This indicates that NPF is either too high or too frequent in the model and this is probably linked to the models having too few larger particles (above $\sim 100$nm) and thus too low coagulation sink.

Furthermore, the underestimation of the larger particles also leads to less available surface area and and too low condensation sink, which may lead to too high H$_2$SO$_4$ and/or SOAG$_{LV}$ concentrations and thus too high nucleation rates.





Our results also go in line with Lee et al. (2013) and Olenius and Riipinen (2017), who show that a higher cut-off diameter leads to over prediction of the aerosol number concentration. They remark that the most likely explanation is the steady-state assumption used in the parameterizations (in our case Lehtinen et al. (2007)). We consider this as the most likely explanation

for the reduction in particles in the modal scheme with OsloAeroSec in our runs as well. In addition, we find that the reduction in number of particles in the modal scheme is largest where the concentration was largest initially, and that in clean, remote regions, there is actually an increase in particle number.

In OsloAeroSec we let more organics ($SOAG_{LV}$ and ELVOC) contribute to growth after 5 nm than is considered in $OsloAero_{def}$ and $OsloAero_{imp}$ (only ELVOC), which is likely why, in the higher atmosphere, OsloAeroSec often produces

more particles than $OsloAero_{imp}$. However, this also illustrate the advantage of a sectional scheme, namely that it is possible to differentiate condensation by particle size.

Related to this, we show that the choice of nucleation parameterization together with the representation of the chemical diurnal variation, has a large influence on the SOA and $H_2SO_4$ contribution to the growth of NPF fraction in particles. This is

especially true when the cutoff diameter is high, as in $OsloAero_{def}$ and $OsloAero_{imp}$. The reason is that the Riccobono et al. (2014) formulation is non-linear, as apposed to the Paasonen et al. (2010) parameterization and thus forms proportionately more particles when $H_2SO_4$ concentrations are high. Including the sectional scheme (OsloAeroSec) counteracts this, both because of particles growing for more than one time step, and that more $SOAG_{LV}$ is allowed to contribute to growth.

In sum these effects illustrate that including NPF in global climate models, often with a very simplified chemistry, should

be done with care. A parameterization may very well be physically sound, but might still give biased results if it is subjected to unrealistic (diurnal) variability concentrations. If the cutoff diameter is high, and the nucleation parameterization has a super-linear relationship with $H_2SO_4$, the influence of organics on survival to larger sizes might be diminished, resulting in a weaker response to changes in BVOC-emissions either in terms of climate feedbacks or forcing for e.g. deforestation/afforestation (Sporre et al., 2019, 2020).

As mentioned, the changes introduced by inserting a sectional scheme are heterogeneous in space and time. The number concentration of modal scheme particles in general decrease where concentrations initially were high and increase where they were low. A topic for further research is therefore how this would influence the modelled effective radiative forcing from cloud-aerosol interaction (ERFaci). If the OsloAeroSec produces more particles in the cleaner pre-industrial atmosphere, and less in the present day atmosphere, it could reduce the ERFaci. Furthermore, the response to both historical and future changes

in BVOC emissions may also be different (Sporre et al., 2020), due to a larger role in the early growth.

Furthermore, considering only the station observation comparison and the general decline in CCN-size number concentration, one might be inclined to think that the same improvement could be achieved by simply reducing the nucleation rate or the survival rate from the (Lehtinen et al., 2007) parameterization. However, the fact that the sectional scheme produces more particles in the remote atmosphere shows that such a quick fix would in fact not produce the same climatic effects and quite

possibly give other sensitivities to emission changes.



Interestingly, the cloud–aerosol effects show clear non-linearities and contradict the simplest assumption that more NPF leads to more CCN which lead to brighter clouds. The correlation between CDNC and NPF particles ($N_{NPF}$, Fig.13) rather show that in polluted regions more NPF results in less CDNC and the reverse in remote regions. This is due to the fact that when NPF increases, the condensate is spread over more particles, reducing the individual particle size so that fewer are activated as CCNs at a given supersaturation (an effect shown in e.g. Sullivan et al. (2018)).

A weakness of the approach of merging a sectional scheme and a modal scheme is that the sectional scheme will grow the particles to size of the volume median diameter of the particles, but when they are inserted into the modal scheme, these particles are represented with a mode distribution, meaning some of them will "shrink" again i.e. be on the lower tail of the distribution. However, this is not uniquely a problem for the sectional scheme – any modal representation of aerosol particles include this effect, the original parameterization in CAM6-Nor makes the same "error". However, improving the early growth parameterization shines a light on this inconsistency – especially because when we plot the size distribution, the number of small particles becomes the sum of the sectional scheme and the modal approximation.

Furthermore, we include a limited number of processes for the sectional scheme (nucleation, coagulation and condensation, while wet/dry deposition are assumed negligible). This is done for simplicity, and is also consistent with the processes considered when using Lehtinen et al. (2007) to parameterize the early growth. Including dry and wet deposition might decrease the number concentrations in the model.

The oxidant concentrations in these simulations are read from monthly mean files and used with a superimposed diurnal variation. Any factor that could impact the oxidant concentration – be it changes in chemical sinks or changes to radiation – will not be accounted for. Since new particle formation is very dependent on this chemistry (see e.g. Lee et al. (2019)), this inhibits how well the model can come to reality.

In terms of computational cost, we tested running one month with standard output fields and the setup described in section 3.1, i.e. active land model and atmosphere, and the computational cost is increased by ∼15 % with OsloAeroSec compared to $OsloAero_{def}$.

## 6 Conclusions

A sectional scheme has been included in the aerosol scheme in CAM6-Nor to explicitly treat the early growth of particles and subsequently feed particles into the pre-existing aerosol scheme. The scheme includes two condensing species, $SOAG_{LV}$ and $H_2SO_4$, and 5 bins. In addition, the diurnal variation in the oxidant concentrations has been improved, and the nucleation parameterization has been updated.

We compare a simulation with the implemented sectional scheme, OsloAeroSec, to two simulations with different versions of the original scheme – one with the default nucleation scheme and oxidant concentrations, $OsloAero_{def}$, and one where these are updated to match the sectional scheme, $OsloAero_{imp}$.

We compare the model output to observations of aerosol concentrations from 2008 and 2009 from 24 stations in Europe (EUSAAR, (Asmi et al., 2011)). We find that all versions of the model overestimates the particles smaller than 100 nm, while



the sectional scheme shows clear improvement compared to the other two. The largest improvements are in the $N_{50-100}$ in the
summer, while changes are insignificant over a 100 nm in diameter.

In general, the sectional scheme reduces the number of particles in the modal scheme near the surface while increasing it
further up in the atmosphere and in remote regions.

The decrease in polluted regions is likely due to overestimation in the original scheme due to the high cut-off diameter of
the aerosol scheme (Olenius and Riipinen, 2017; Lee et al., 2013).

The relative contributions of $H_2SO_4$ and $SOAG_{LV}$ to the early growth of the particles changes between all the model
versions. This is due to the complex interplay between in the introduction of diurnal variation of the oxidants, changes to the
nucleation equation and the introduction of a sectional scheme. This illustrates that care must be taken when implementing
NPF in global models because a highly simplified chemistry may have unintended effects on the sensitivities of NPF to e.g.
changing emissions emissions.

We also analyse the cloud changes and show how the effect of the changes in NPF are heterogeneous in space. An assumption
that more particles from NPF leads to more activated CCN and increased CDNC fails in most regions close to the surface, where
the inverse is true. Higher up in the atmosphere and in remote regions however, the relationship holds.

In general, this study shows that combining a sectional scheme for early growth with a modal scheme for the larger particles
is both possible and that this treatment of early growth improves the representation in the smaller parts of CCN size range.

*Code and data availability.* The output data from the simulations used are available for download at https://doi.org/10.11582/2020.00056
(Blichner, 2020a). The model code of NorESM2, release 2.0.1, is available at https://doi.org/10.5281/zenodo.3760870 (Seland et al., 2020a).
The code modifications in OsloAeroSec, simulation configurations and setup instructions are released at https://doi.org/10.5281/zenodo.
4265057 (Blichner, 2020b). The postprocessing code used to produce the figures are available at https://doi.org/10.5281/zenodo.4265033
(Blichner, 2020c).

*Author contributions.* SMB did the model code development and performed the simulations with NorESM. SMB did the data analysis and
wrote the manuscript. RM and SMB contributed with the idea. SMB, MKS and TKB contributed with discussions regarding the experimental
design and data analysis. All contributors have contributed to the discussions regarding the manuscript.

*Competing interests.* No competing interests.

*Acknowledgements.* This work was funded under the LATICE strategic research initiative funded by the Faculty of Mathematics and Natural
Sciences at the University of Oslo. This work has been financed by the Research Council of Norway (RCN) through the NOTUR/Norstore





project NN2806K and NS9066K. We would like to thank the EUSAAR project for use of the measurements. Thanks to Diego Aliaga for help and discussions on the data analysis and visualization. Inger Helene Karset for scientific discussions.





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





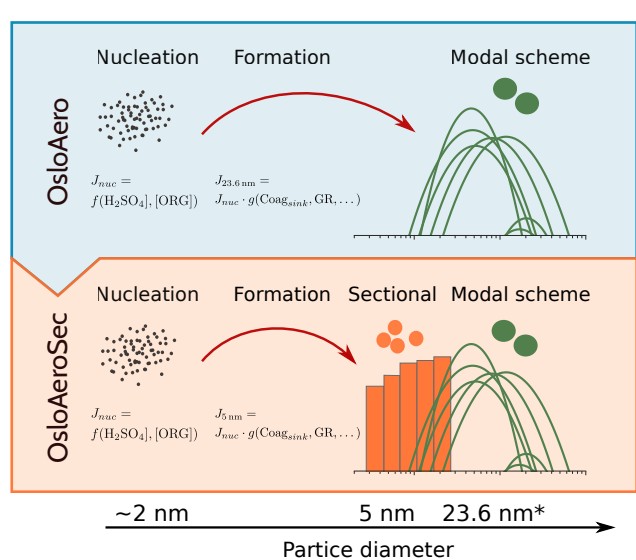

**Figure 1.** Illustration of changes from OsloAero to OsloAeroSec. In both versions, the nucleation rate is calculated at around 2 nm followed by a calculation of the formation rate (the particles surviving) at 5 nm and 23.6 nm in OsloAeroSec and OsloAero respectively, with Lehtinen et al. (2007). In OsloAero, these particles are inserted directly into the modal scheme, while in OsloAeroSec, the particles are inserted into the sectional scheme where they can be affected by growth and coagulation over time and space. Finally, the particles in the sectional scheme are moved from the last bin of the sectional scheme to the modal scheme. *23.6 nm is the number median diameter of the mode the particles from the sectional scheme are moved to, but particles are actually grown to the volume median diameter before they are moved to the modal scheme in order to conserve mass.





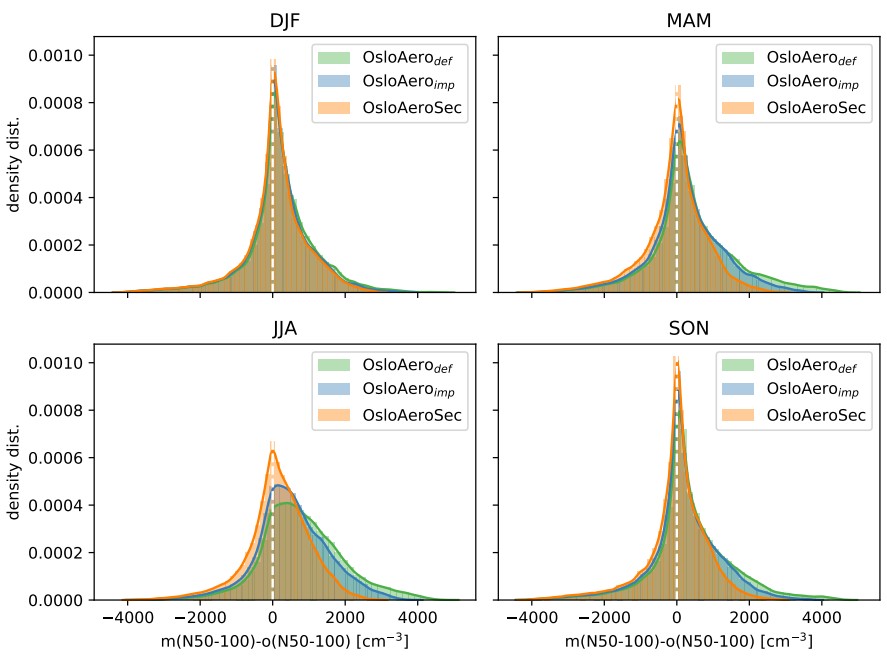

**Figure 2.** Seasonal distribution of modelled $N_{50-100}$ minus observed $N_{50-100}$ for all EUSAAR stations (Asmi et al., 2011). We use hourly resolution and all available station data is is included.





**Figure 3.** Histogram of modelled $N_{50-100}$ minus observed $N_{50-100}$ for each season and EUSAAR station (Asmi et al., 2011). We use hourly resolution and all available station data is is included. Zeppelin (ZEP), Mace Head (MHD), Aspvreten (ASP), SMEAR II (SMR), Pallas (PAL), Kosetice (OBK), Vavihill (VHL), Melpitz (MPZ), Waldhof (WAL), Bösel (BOS), Hohenpeissenberg (HPB), K-Puszta (KPO), JRC-Ispra (JRC), Finokalia (FKL), Jungfraujoch (JFJ), Schauinsland (SSL), Zugspitze (ZSF), Monte Cimone (CMN), BEO Moussala (BEO), Puy de Dôme (PDD) Preila (PLA), Birkenes b (BIR), Harwell (HWL), Cabauw (CBW).



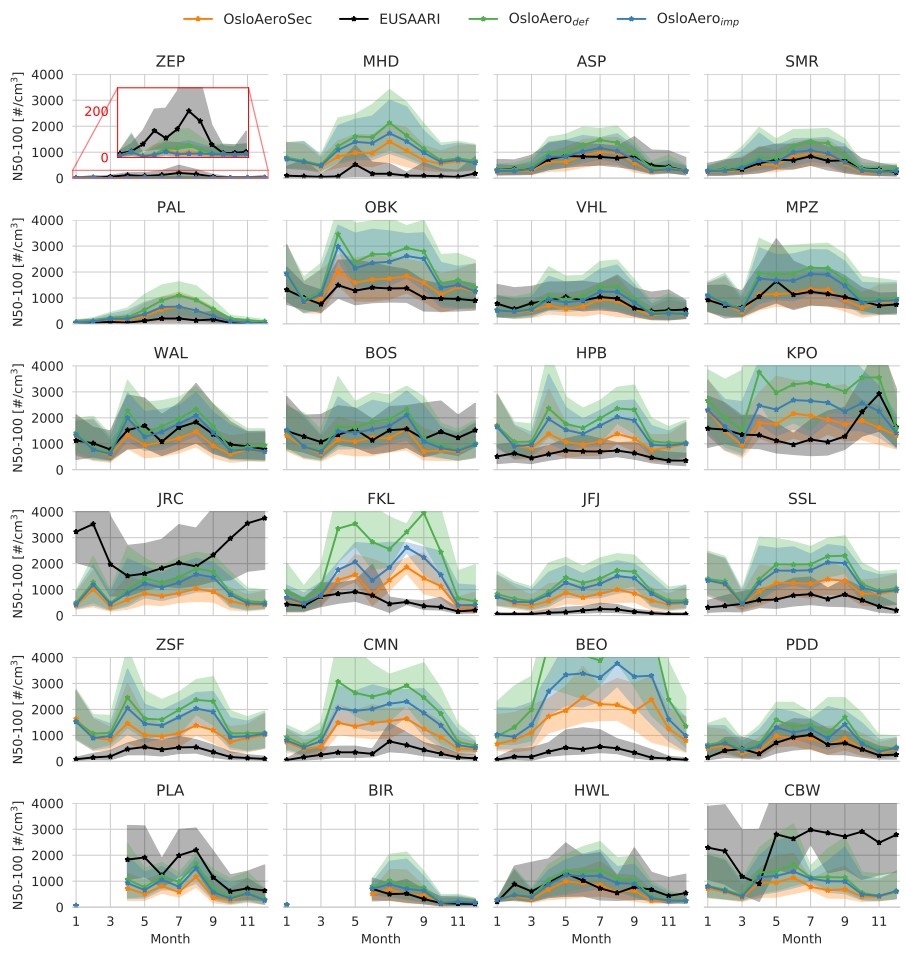

**Figure 4.** $N_{50-100}$ monthly median (solid line) and percentiles (shaded, 16th to 84th) for each station for each model version and the observed values (Asmi et al., 2011). Stations where the full graph is not shown due to the axis limits are shown in full in Fig. S1.





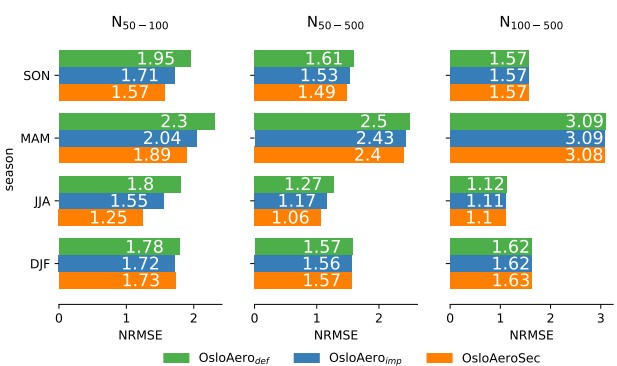

**Figure 5.** Normalized root mean square error (NRMSE) for each season and each model version compared to the EUSAAR dataset (Asmi et al., 2011). The root mean square error is normalized by the mean of the observed values.



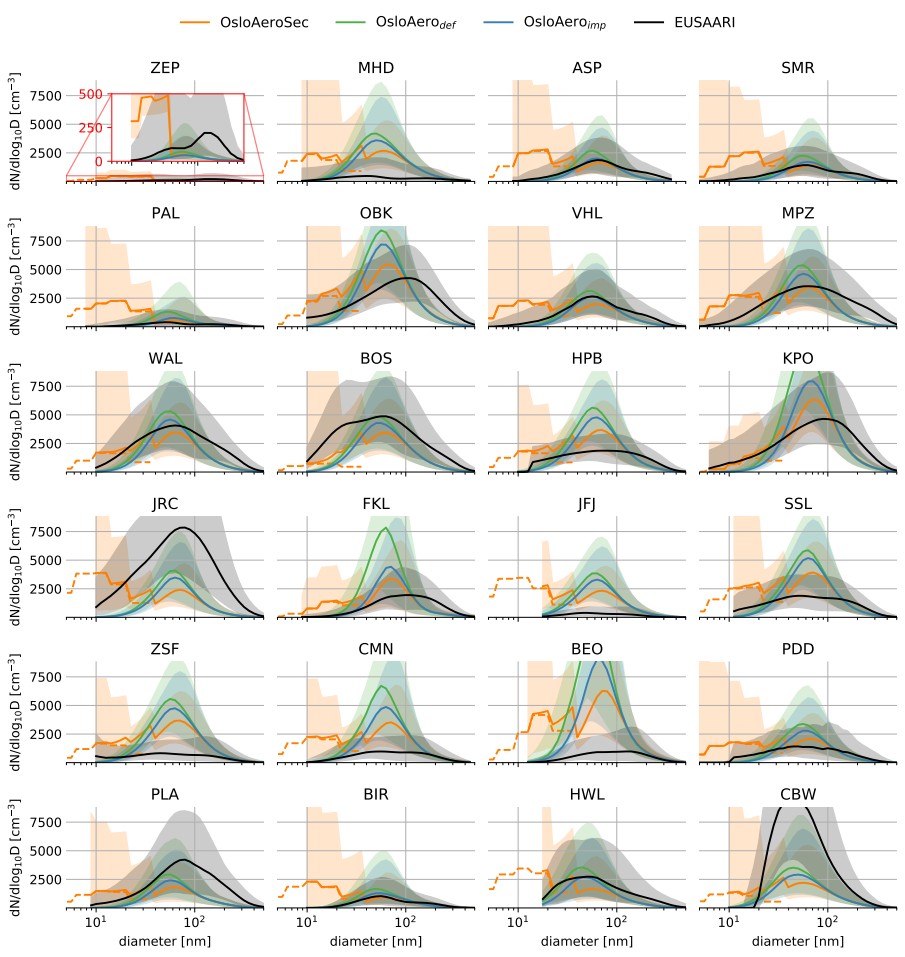

**Figure 6.** Median (solid line) particle number size distribution and shading from 16th to 84th percentiles for observations (Asmi et al., 2011) and models. All data when and where observations are available is included.





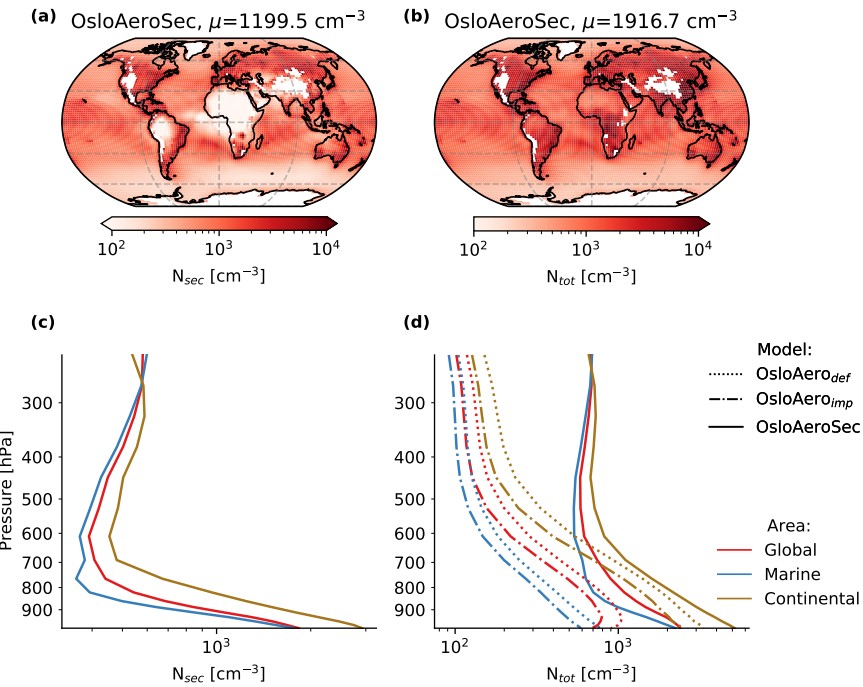

**Figure 7.** Modelled particle number concentrations. The top panels shows maps near-surface average concentrations for $N_{sec}$ (a) and $N_{tot}$ (b) in OsloAeroSec. The bottom panels show average profiles globally, over continents (continental) and over ocean (marine) for $N_{sec}$ (c) and $N_{tot}$ (d). In d, OsloAero$_{def}$ and OsloAero$_{imp}$ are also included.





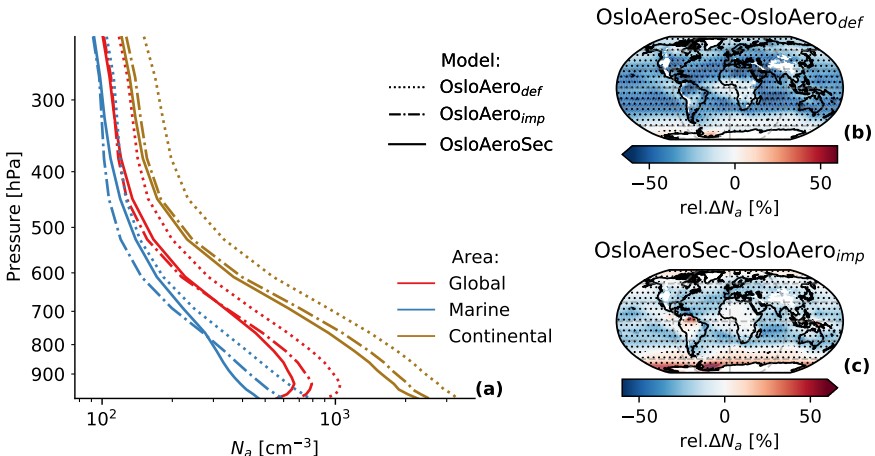

**Figure 8.** Comparison of $N_a$ from OsloAero$_{def}$ and OsloAero$_{imp}$ to OsloAeroSec. Panel a shows profiles for mean of regions (global, marine and continental) for the model versions. Panel b and c show the relative difference in near-surface mean of OsloAeroSec to OsloAero$_{def}$ and OsloAero$_{imp}$, respectively. Areas where the difference is significant (95%) are marked with dots.



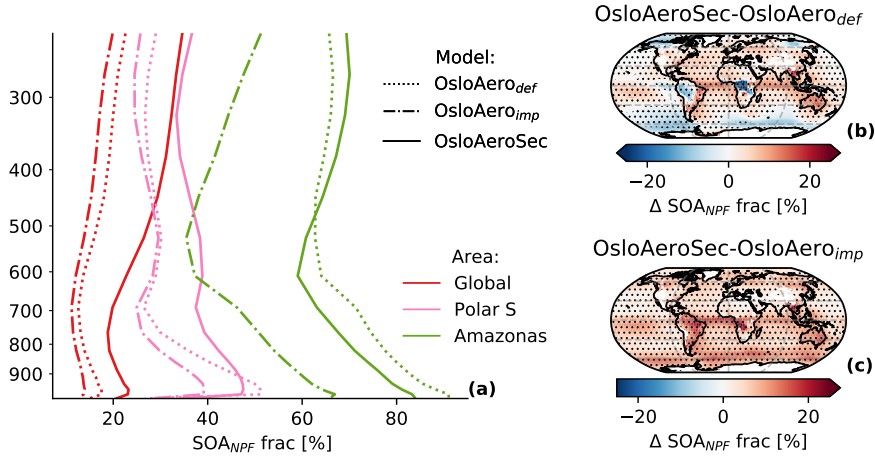

**Figure 9.** The SOA fraction of $N_{NPF}$ mass ($SOA_{NPF}$), i.e. the fraction of the growth of the particles before they reach the modal scheme which is due to organics. Panel a shows profiles for regions (Global, Polar S(outh), Amazonas) with each model. Panels b and c show the difference in near-surface mean values for OsloAeroSec minus OsloAero$_{def}$ and OsloAero$_{imp}$ (c), respectively. Areas where the difference is significant (95%) are marked with dots.



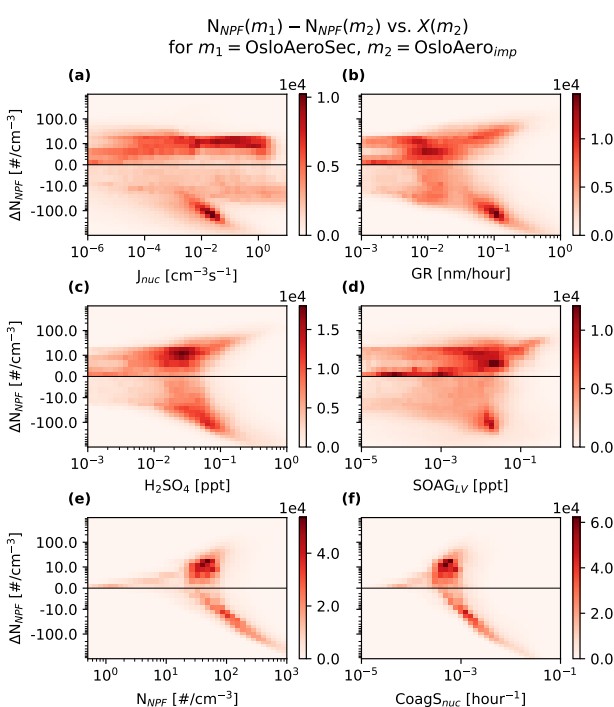

**Figure 10.** Two-dimensional histogram of the relation between various factors in the original model run OsloAero$_{imp}$, and the change in number of particles from NPF, N$_{NPF}$ between OsloAeroSec and OsloAero$_{imp}$. The color shows the number of model grid cells which fall within the x,y-range using monthly mean files. Only grid cells below 100 hPa are included. The values on the x-scale are the nucleation rate (a), the growth rate of newly formed particles (b), the mixing ratio of H$_2$SO$_4$ (c), the mixing ratio of SOAG$_{LV}$ (d), the concentration of particles from NPF (e) and the coagulation sink for newly formed particles (f). See Fig. S12 for the same plot, but with N$_{NPF}$ from OsloAero$_{imp}$, i.e. not the change.



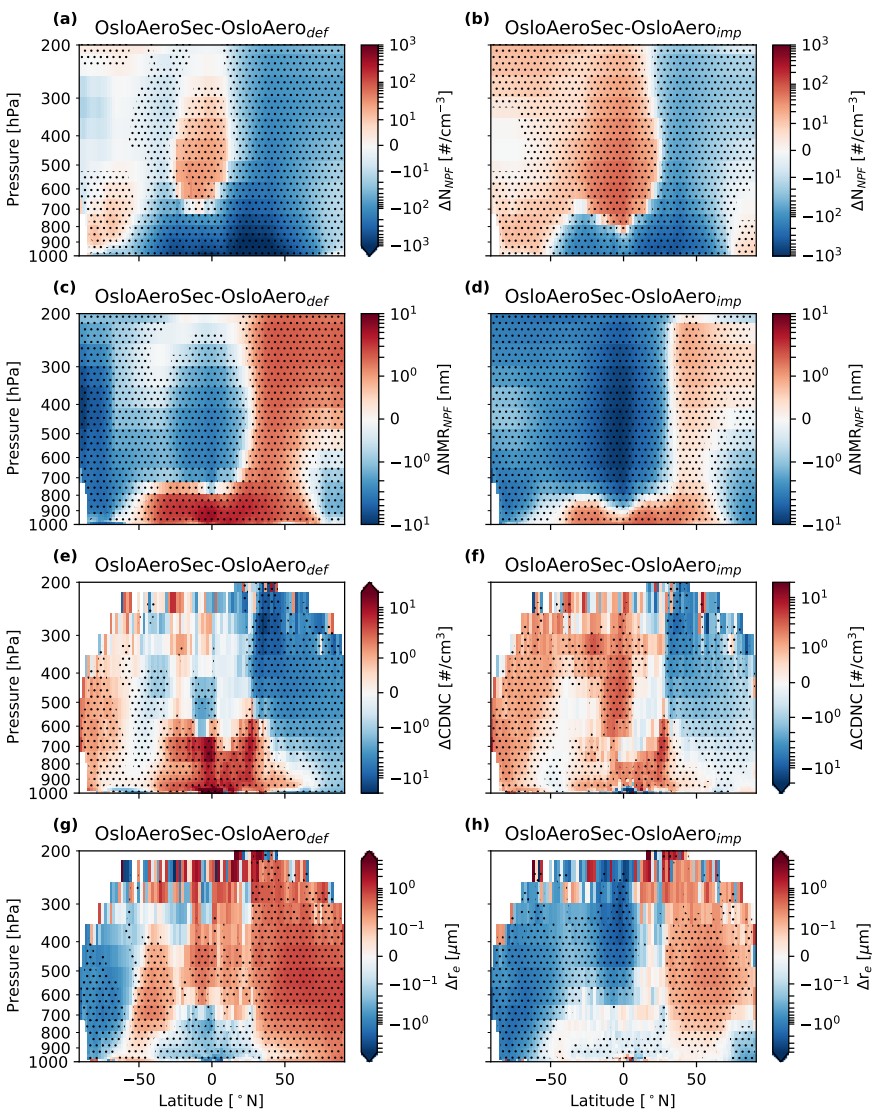

**Figure 11.** Zonally averaged change between OsloAeroSec and OsloAero$_{def}$ (left column) and OsloAeroSec and OsloAero$_{imp}$ (right column) in in N$_{NPF}$ (a and b), number median radius for NPF-particles (NMR$_{NPF}$, c and d), cloud droplet number concentration (CDNC, e and f) and cloud drop number concentration ($r_e$, g and h). Areas where the difference is significant (95%) are marked with dots.



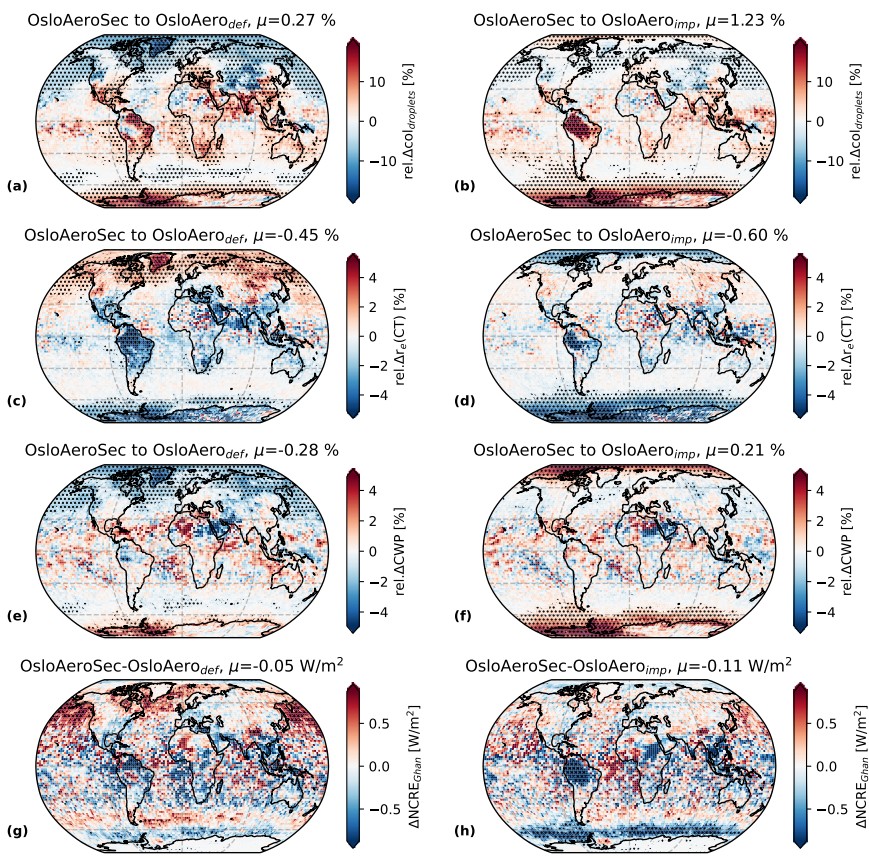

**Figure 12.** Changes to cloud properties. The left column shows the difference between OsloAeroSec and OsloAero$_{def}$ and the right column shows the difference between OsloAeroSec and OsloAero$_{imp}$. Panels a and b show the relative difference in cloud top droplet number concentration (CDNC(CT)), panels c and d show the relative difference in effective droplet radius at cloud top ($r_r$(CT)), panels e and f show the relative difference in cloud water path (CWP) and finally panel g and h show the difference in net cloud radiative effect (NCRE) calculated as recommended in Ghan (2013). Areas where the difference is significant (95%) are marked with dots.



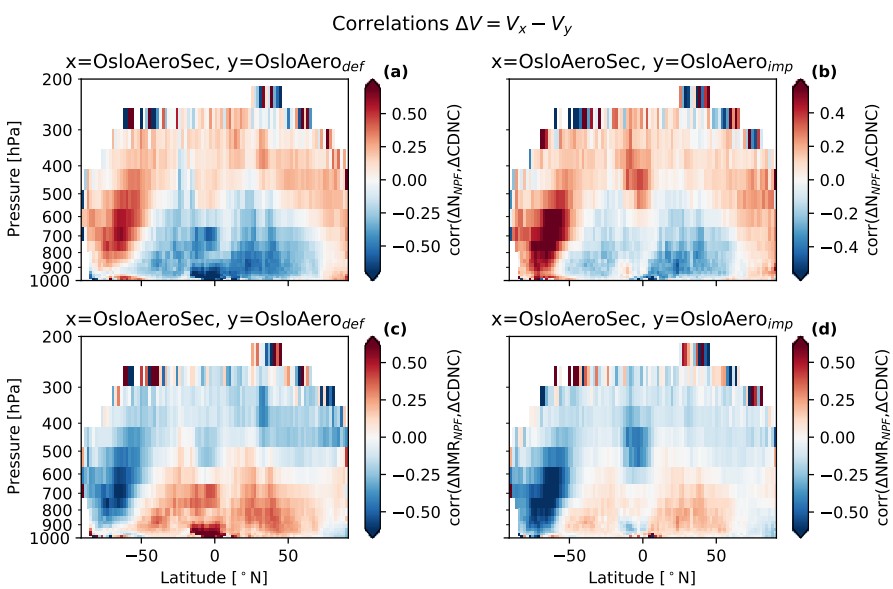

**Figure 13.** Correlations between the change in CDNC and $N_{NPF}$ (top) and number median radius of the NPF particles ($NMR_{NPF}$)(bottom). Plots on the left side are for the difference OsloAeroSec − OsloAero$_{def}$ ($\Delta V = V_{\text{OsloAeroSec}} - V_{\text{OsloAero}_{def}}$ for variable $V$) and plots on the right are for OsloAeroSec − OsloAero$_{imp}$.





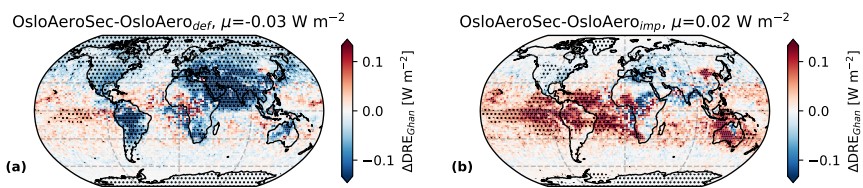

**Figure 14.** Change in direct aerosol effect for OsloAeroSec minus OsloAero$_{def}$ (a) and OsloAeroSec minus OsloAero$_{imp}$ (b). The direct radiative effect is calculated as recommended by Ghan (2013). Areas where the difference is significant (95 %) are marked with dots.



**Table 1.** Simulation overview. See detailed description in section 3.

| Simulation | Nucleation parameterization | Oxidant treatment | Early growth treatment |
|---|---|---|---|
| OsloAeroSec | $A_3[H_2SO_4]^2 \times [ELVOC]$ * | Improved diurnal variation | Lehtinen et al. (2007) + sectional scheme |
| OsloAero$_{imp}$ | $A_3[H_2SO_4]^2 \times [ELVOC]$ * | Improved diurnal variation | Lehtinen et al. (2007) |
| OsloAero$_{def}$ | $A_1[H_2SO_4] + A_2[ELVOC]$ † | Default diurnal variation | Lehtinen et al. (2007) |

$A_1 = 6.1 \times 10^{-7}$ s$^{-1}$

$A_2 = 3.9 \times 10^{-8}$ s$^{-1}$

$A_3 = 3.27 \times 10^{-21}$ cm$^6$s$^{-1}$

* Riccobono et al. (2014)

† Paasonen et al. (2010)





**Table 2.** Region overview. These regions are used to create vertical average profiles.

| Region name | Description | Latitudes | Longitudes |
|---|---|---|---|
| Continental | Grid boxes with >50% land | | |
| Marine | Grid boxes with <50% land | | |
| Global | | | |
| Polar N | | 66.5 – 90 °N | 180 °W – 180 °E |
| Polar S | | 66.5 – 90 °S | 180 °W – 180 °E |
| Amazonas | | 16 °S – 2 °N | 74 – 50 °W |





**Table 3.** Model variable definitions.

| Variable name | Definition |
| --- | --- |
| $N_a$ | Number of particles excluding those in the sectional scheme |
| $N_{tot}$ | Number of particles including those in the sectional scheme |
| $N_{sec}$ | Number of particles in the sectional scheme |
| $N_{\text{NPF}}$ | Number of particles from NPF excluding those in the sectional scheme |
| $N_{d_1-d_2}$ | Number of particles with diameter $d$ such that $d_1 \leq d \leq d_2$ |
| $N_{d_1}$ | Number of particles with diameter $d$ such that $d_1 \leq d$ |