# Peer review of "Implementing a sectional scheme for early aerosol growth from new particle formation in the Norwegian Earth System Model v2: comparison to observations and climate impacts"

_Geoscientific Model Development, 2020_

## Referee Comment (RC1) · Anonymous Referee #1 · 6 Dec 2020

**Review of Blichner et al, Implementing a sectional scheme for early aerosol growth from new particle formation in the Norwegian Earth System Model v2: comparison to observations and climate impacts**

This paper documents a new sectional sub-model for representing the evolution of nucleated particles in NorESM. It is a good paper: interesting, well-written and well within the scope of GMD. I have two major comments and a few minor suggestions I would like the authors to address before the paper is published.

**Major comments:**
Often, new aerosol microphysics schemes have been tested in box modeling frameworks before implementing into a global model, but such tests are not reported here. In my opinion box modeling would address my two major comments most effectively and might also shed light on other behavior of the model.

(Line 270) Have you considered the effects of numerical diffusion (Jacobson 2005, section 13.5.3)? To minimize these wouldn't a moving-center structure be better, as done in the sectional microphysics models TOMAS (Adams et al, JGR 2002), GLOMAP-bin (Spracklen et al, ACP 2005) though not some others (e.g. APM, Yu et al ACP 2009, I think)? Is it possible to show that numerical diffusion doesn't matter?

What is the timestep and is the same timestep used for advection and for microphysics? Is it half an hour for both as implied on line 106? Are the results sensitive to the microphysics timestep (or the model timestep, if it's the same)? (Maybe add timestep in section 2.1 introduction, since it is needed to understand 2.1.2). My understanding is that ~30 minutes is quite a long timestep for nucleation-mode aerosol microphysical processes unless sulfuric acid is in pseudo-steady state (see Pierce & Adams, Aerosol Science and Technology 2008), though I assume it is fine for models that don't resolve the nucleation mode like OsloAero_def. For example, in GLOMAP where sulfuric acid is not in pseudo-steady-state, the timestep is 3 minutes for condensation, nucleation and coagulation (Mann et al GMD 2010), while in APM I think the timestep is variable but also less than 30 minutes. So I would expect your results might change in some situations if (say) a 5 or 10 minute timestep were used. I note there is a helpful discussion of some timing issues at lines 455-460, but sensitivity studies are called for here.

**Minor comments**:

The detailed comparison at surface sites is very nice but of course it only presents a two-dimensional picture. Entrainment of the nucleated particles you are interested in from the upper troposphere is likely a bigger source of cloud-level CCN than boundary-layer new particle formation (e.g. Merikanto et al, ACP 2009). So it would be really nice to see a comparison of Ntot to aircraft measurements e.g. from ATom (e.g. Williamson et al, Nature 2019; Ranjithkumar et al, ACPD 2020), but I realise this is a lot to ask. Perhaps something to think about for a follow-up paper?

Line 117: Would be nice to be consistent in referring only to diameters, not radii, except where necessary. Your scheme accounts for 5-39.6nm diameter particles; the smallest mode apparently has an initial diameter of 47.2nm. Or is 23.6nm actually the diameter, as stated in the caption of Figure 1? If not, what happens between 39.6nm and 47.2nm?

Line 199: would be helpful to clarify that this is the original unperturbed version of the model

Section 2.2.1 Would be worth commenting here that the Riccobono et al parameterization of nucleation is still very uncertain, as it lacks a temperature dependence and was not based on ELVOC measurements, but rather on pinanediol ELVOC precursors, and it doesn't take account of the role of ammonia in nucleation.

Section 2.3 Just a comment: I imagine this is a big improvement, and a good catch that could be implemented in the default NorESM for all users, even those not prepared to accept the extra cost of your nucleation-mode microphysics scheme, so the OsloAeroImp simulation is useful.

Section 3: Is 1.9x2.5 the resolution of the whole model, or just the SST/sea ice input file? If the latter, what is the model resolution?

Line 310 "from 2007 to 2014 inclusive" perhaps?

Line 395: what is the "tail one NPF-particle mode"? Also please don't start a new paragraph with "This" - it makes the text hard to follow. I don't really understand the content of this paragraph. Is there no way to transfer particles from one mode to another if the mode gets too large or small? This seems inconsistent with previous descriptions, somewhere you also suggest particles in the lowest mode could shrink?

Line 405 "the sectional scheme overestimated the number of particles for this"
Can you specify "the number of particles at low diameters"
Line 407 "the surface distribution"-> "the distribution of particle surface areas"

Line 410: I don't think you represent nitrate or ammonium aerosol, could this, or errors in biogenic or anthropogenic SOA, be responsible for an underestimated condensation sink?

Line 415: should probably specify the size range over which you reduce concentrations.

Line 430: maybe comment that the large change in the upper troposphere is expected because the default model versions simply don't represent small particles.

Figure 6: It seems a shame to cut off the y axes at about 8000 – why not 12000 for example?

Figure 11: I think e and f are effective radius, not droplet concentration.

You might consider reducing the total number of figures; they are many for a paper whose text is relatively short. Figure 3, for example, does not seem to convey very different information to Figure 4, and Figure 5 has much the same information as Figure 2, given that the changes in 100-500nm particle concentrations between simulations are negligible. On the other hand, I found Figure S6 illuminating even though it is similar to Figure 6; Figure S6 might be worth promoting to the main text.

---

## Referee Comment (RC2) · Anonymous Referee #2 · 18 Dec 2020

The manuscript "Implementing a sectional scheme for early aerosol growth from new particle formation in the Norwegian Earth System Model v2: comparison to observations and climate impacts" introduces a new method for calculating the formation and growth of newly formed particles. The method applies the sectional approach for describing the smallest particles together with the modal method for the larger particles. The paper is well and clearly written and is definitely in the scope of Geoscientific Model Development. The method presented here is definitely relevant for the global aerosol modelling community and it can be used in any modal aerosol model. I can

recommend publishing the manuscript after the following issues are addressed:

- The confidence in the improved model could be strengthened by demonstrating its performance in a 0-dimensional framework comparing it to for example a high resolution sectional model.

- The differences between different model version are discussed from the view point of aerosol microphysics. However, for example in-cloud impact scavenging affects strongly the number of small particles. I am not familiar with the size dependence in CAM-Nor wet deposition scheme; can wet scavenging cause differences between the simulated aerosol concentrations in different model versions?

- On Page 17, Line 510 onward, the explanation between the different behavior is slightly ambiguous, especially the last sentence of the paragraph: "Additionally, in highly polluted areas, an increased number of particles may inhibit activation because more particles compete for the same water vapor". Does it want to say that in polluted areas, newly formed particles decrease supersaturation so much that it suppresses activation? If so, my understanding that in order to suppress the number of activated particles by increasing the number of particles, number concentrations have to be extremely high and would occur in very few parts of the atmosphere.

  Does also the modal representation affect this behavior? With a high number of NPF particles entering the smallest mode, the mode will shift to smaller sizes artificially suppressing activation.

**Technical comments:**

- Page 2, Line 49: Correct "depend" to "dependent"

- The model is referred to as CAM6-Nor and CAM-Nor6. On Page 9, Line 256, CAM6-Oslo is mentioned. Please correct the wrong namings.

---

## Author Comment (AC1) · 16 Mar 2021

**Authors response**

Sara M. Blichner[1], Moa K. Sporre[2], Risto Makkonen[3,4], and Terje K. Berntsen[1]

[1]Departmenet of Geosciences and Centre for Biogeochemistry in the Anthroposcene, University of Oslo, Oslo, Norway
[2]Department of Physics, Lund University, Lund, Sweden
[3]Institute for Atmospheric and Earth System Research / Physics, Faculty of Science, University of Helsinki, Finland
[4]Climate System Research, Finnish Meteorological Institute, Helsinki, Finland

**Correspondence:** Sara Marie Blichner (s.m.blichner@geo.uio.no)

We would like to thank the reviewers for very helpful and insightful comments which have improved this manuscript. We present our responses to the questions and comments below. The comments from the reviewer are written first, and our responses follow in italic. We will answer the comments from both referees in this document.

**Referee 1**

**Major comments**

Often, new aerosol microphysics schemes have been tested in box modeling frameworks before implementing into a global model, but such tests are not reported here. In my opinion box modeling would address my two major comments most effectively and might also shed light on other behavior of the model.

Have you considered the effects of numerical diffusion (Jacobson 2005, section 13.5.3)? To minimize these wouldn a moving-center structure be better, as done in the sectional microphysics models TOMAS (Adams et al, JGR 2002), GLOMAP-bin (Spracklen et al, ACP 2005) though not some others (e.g. APM, Yu et al ACP 2009, I think)? Is it possible to show that numerical diffusion doesn't matter?

- *This is an interesting point. Because this study is done for an ESM and one necessary requirement was to keep the computational costs down, we chose the quasi-stationary structure instead of the moving-center structure, due to it needing fewer tracers. The downside to this is that it is more prone to numerical diffusion.*
  *While we would have liked to have a box model, unfortunately there is none available for OsloAero. It would be a nice addition to the model though. However, we have now done some sensitivity tests where we vary the number of bins and timesteps, which indicates the strength of the numerical diffusion. This is added to the manuscript in a new section 4.3 and in the supplementary. When the number of bins is increased from 5 to 8 bins, the number of NPF particles increases with approximately 7.5 %, while reducing the number of bins to 3 gives a reduction in particles of approximately 12 %. This indicates that numerical diffusion does play a role, but that the effect is much smaller than the change between the default scheme and the new scheme.*

What is the timestep and is the same timestep used for advection and for microphysics? Is it half an hour for both as implied on line 106? Are the results sensitive to the microphysics timestep (or the model timestep, if it's the same)? (Maybe add timestep in section 2.1 introduction, since it is needed to understand 2.1.2). My understanding is that 30 minutes is quite a long timestep for nucleation-mode aerosol microphysical processes unless sulfuric acid is in pseudo-steady state (see Pierce & Adams, Aerosol Science and Technology 2008), though I assume it is fine for models that don't resolve the nucleation mode like OsloAero_def. For example, in GLOMAP where sulfuric acid is not in pseudo-steady-state, the timestep is 3 minutes for condensation, nucleation and coagulation (Mann et al GMD 2010), while in APM I think the timestep is variable but also less than 30 minutes. So I would expect your results might change in some situations if (say) a 5 or 10 minute timestep were used. I note there is a helpful discussion of some timing issues at lines 455–460, but sensitivity studies are called for here.

– *This is a very good point and we have now carried a sensitivity test on this and added the information to the paper. The timestep is half an hour both for advection and microphysics, though if the condensational growth causes the particles to skip one bin, the time step is halved locally for the condensation/nucleation code. We have added a sentence on this in the model description at the end of section 2.2.2:*

*"The time step within the nucleation and condensation code is locally divided in two compared to the rest of the code (thus 15 minutes), and if the particles in the sectional scheme grow fast enough to skip a bin, the time step is further divided in two until it is small enough."*

*Additionally, we have performed sensitivity tests (added to the supplementary) where we have tested dividing the timestep locally for condensation, coagulation and nucleation in 10, giving a timestep of only 3 minutes. The change in particle concentration from this is less than 3 %, which is encouraging. This may be partially because the scheme starts at 5 nm instead of at a smaller diameter. As mentioned above, we have added a section in the results and the supplementary material.*

**Minor comments**

The detailed comparison at surface sites is very nice but of course it only presents a two dimensional picture. Entrainment of the nucleated particles you are interested in from the upper troposphere is likely a bigger source of cloud-level CCN than boundary-layer new particle formation (e.g. Merikanto et al, ACP 2009). So it would be really nice to see a comparison of Ntot to aircraft measurements e.g. from ATom (e.g. Williamson et al, Nature 2019; Ranjithkumar et al, ACPD 2020), but I realise this is a lot to ask. Perhaps something to think about for a follow-up paper?

– *Indeed, this is a very nice idea for a follow up paper.*

Line 117: Would be nice to be consistent in referring only to diameters, not radii, except where necessary. Your scheme accounts for 5–39.6nm diameter particles; the smallest mode apparently has an initial diameter of 47.2nm. Or is 23.6nm actually the diameter, as stated in the caption of Figure 1? If not, what happens between 39.6nm and 47.2nm?

– *This is a typo and it is supposed to read 23.6 nm in diameter. Line 117 has been corrected and now reads:*

*"In the original modal scheme in NorESM, the smallest mode has an initial number median diameter of 23.6 nm (volume*

*median diameter of 39.6 nm). [. . . ]"*

*We have also substituted radius for diameter in one other place (line 124).*

*We have further added some clarifications in the manuscript, e.g. in the figure text of figure 1, we have added 39.6 nm as the volume median diameter:*

60     *"Finally, the particles in the sectional scheme are moved from the last bin of the sectional scheme to the modal scheme. \*23.6 nm is the number median diameter of the mode the particles from the sectional scheme are moved to, but particles are actually grown to the volume median diameter (39.6 nm) before they are moved to the modal scheme in order to conserve mass."*

*Explanation: The initial diameter prior to growth is 23.6 nm. This is the median diameter of the "background mode", that*

65     *is prior to any condensational growth. OsloAero is set up in a way such that the condensate tracers (process tracers) are added on top of these background tracers, then a look-up table approach is used to determine the size distribution and the diameter 47.2 is then a log normal fitting to the resulting distribution after growth. This is somewhat complicated, and explained in more detail in OsloAero description papers (see e.g. Kirkevag 2018). The particles are added at 39.6 nm which is the volume median diameter, so as to conserve mass and number. This is described on line 236–240, but we*

70     *agree that it was confusing the way the introduction was written. We hope that these adjustments will suffice.*

Line 199: would be helpful to clarify that this is the original unperturbed version of the model.

    – *This is in fact described at the beginning of section 2:*

    *"We start by briefly describing the Norwegian Earth System Model (NorESM) in general before giving a detailed description of its aerosol model, OsloAero, in section 2.1. After this in section 2.2, we will describe what changes to said*

75     *aerosol scheme that have been introduced in OsloAeroSec. In general, the aerosol scheme after NPF and early growth is left as it is."*

    *We agree that it could be made clearer though and have changed this text to:*

    *"We start by briefly describing the Norwegian Earth System Model (NorESM) in general before giving a detailed description of its **default** aerosol model, OsloAero, in section 2.1. [. . . ]"*

80     *We have also added a sentence at the end of line 199: "This is the default nucleation equation in OsloAero and is changed in OsloAeroSec – see section 2.2.1."*

Section 2.2.1 Would be worth commenting here that the Riccobono et al parameterization of nucleation is still very uncertain, as it lacks a temperature dependence and was not based on ELVOC measurements, but rather on pinanediol ELVOC precursors, and it doesn't take account of the role of ammonia in nucleation.

85     – *We add the following to the text:*

    *"Note that even though it is likely that the Riccobono et al. (2014) parameterization represents an improvement compared to Paasonen et al. (2010), large uncertainties remain due to the fact that the Riccobono et al. (2014) parameterization was developed based on an ELVOC precursor (pinanediol), rather than actual ELVOC measurements, and that it does*

*not take into account other factors that have been shown to be of importance, like temperature and ammonia (see e.g.*
*Semeniuk and Dastoor, 2018)."*

Section 2.3 Just a comment: I imagine this is a big improvement, and a good catch that could be implemented in the default
NorESM for all users, even those not prepared to accept the extra cost of your nucleation-mode microphysics scheme, so the
OsloAeroImp simulation is useful.

  – *Thanks for this comment! We hope that this improvement will be implemented into the default NorESM version.*

Section 3: Is 1.9x2.5 the resolution of the whole model, or just the SST/sea ice input file? If the latter, what is the model
resolution?

  – *Good point. We have added the following sentence in section 3, approximately at line 313:*
    *"We use a 1.9° (latitude) x 2.5° (longitude) resolution grid with 32 height levels from the surface to  2.2 hPa in hybrid*
    *sigma coordinates."*

Line 310 "from 2007 to 2014 inclusive" perhaps?

  – *Good suggestion, we have changed the manuscript accordingly.*

Line 395: what is the "tail one NPF-particle mode"? Also please don't start a new paragraph with "This" - it makes the text
hard to follow. I don't really understand the content of this paragraph. Is there no way to transfer particles from one mode to
another if the mode gets too large or small? This seems inconsistent with previous descriptions, somewhere you also suggest
particles in the lowest mode could shrink?

  – *Apologies, there was not supposed to be a paragraph shift here. The way OsloAero is set up, the particles may not be*
    *transferred from one mode to the next by condensational growth. We cannot see that we have suggested otherwise in the*
    *text. Any shrinking would be in reference to the median diameter of the mode. To clarify, we have changed as following:*
    *"The greatest improvement is seen in N50-500 and in summer, followed by SON and MAM, while DJF is mostly un-*
    *changed. N50-500 shows improvement in the same seasons, while there only small improvements in prediction skill for*
    *N100-500.*
    *The lack of change in prediction skill for particles larger than 100 nm likely originates from the fact that in CAM6-Oslo,*
    *the NPF particles do not change mode by condensational growth – rather the whole mode grows in number median diam-*
    *eter. Thus, the variability in concentrations of particles larger than 100 nm is dominated by primary particle emissions,*
    *which we do not alter here."*

Line 405 "the sectional scheme overestimated the number of particles for this" Can you specify "the number of particles at low
diameters"

  – *We agree and have changed this to: "One important reason why the sectional scheme overestimated the number of the*
    *particles at the smallest sizes may be that the number of particles above 100 nm [. . . ]"*

120   Line 407 "the surface distribution"-> "the distribution of particle surface areas"

     – *We agree and have changed the text accordingly.*

Line 410: I don't think you represent nitrate or ammonium aerosol, could this, or errors in biogenic or anthropogenic SOA, be responsible for an underestimated condensation sink?

     – *This is a good point and could be a good subject for a future study. At this point, there may be many reasons for the*
125       *underestimation. However, a detailed investigation of this would be beyond the scope of this study.*

Line 415: should probably specify the size range over which you reduce concentrations.

     – *Good point, we have changed accordingly: "We especially get a reduction in number concentrations of diameters above*
       *50 nm where they are significantly overestimated."*

Line 430: maybe comment that the large change in the upper troposphere is expected because the default model versions simply
130   don't represent small particles.

     – *Yes, good point, we have added: "[...] where Ntot is very low in OsloAero_imp and OsloAero_def because the smallest*
       *particles are simply not represented in these model versions".*

Figure 6: It seems a shame to cut off the y axes at about 8000 – why not 12000 for example?

     – *We agree and have adjusted the figure accordingly.*

135   Figure 11: I think e and f are effective radius, not droplet concentration.

     – *There seems to be a typo in the text. It read "f) and cloud drop number concentration (re , g and h)." but should read "f)*
       *and cloud droplet effective radius (re , g and h)." This has been corrected.*

You might consider reducing the total number of figures; they are many for a paper whose text is relatively short. Figure 3, for example, does not seem to convey very different information to Figure 4, and Figure 5 has much the same information as
140   Figure 2, given that the changes in 100–500nm particle concentrations between simulations are negligible. On the other hand, I found Figure S6 illuminating even though it is similar to Figure 6; Figure S6 might be worth promoting to the main text.

     – We have moved figure 3 and figure 5 to the supplementary.

**Referee 2**

The confidence in the improved model could be strengthened by demonstrating its performance in a 0-dimensional framework
145   comparing it to for example a high resolution sectional model.

     – *We agree with this comment. However, the OsloAero model does not have a box model setup. It would be a nice addition*
       *to the model though. See also replies to referee 1, first comment, above.*

The differences between different model version are discussed from the view point of aerosol microphysics. However, for example in-cloud impact scavenging affects strongly the number of small particles. I am not familiar with the size dependence in CAM-Nor wet deposition scheme; can wet scavenging cause differences between the simulated aerosol concentrations in different model versions?

— *We are not entirely sure what size range is meant by "small particles" here. As mentioned in the paper, the particles in the sectional scheme do not interact with clouds, and thus it would not play a role here. For the slightly larger particles which are already in the modal scheme, their impact scavenging does depend on the diameter. However, in the current setup the diameter used in this particular calculation is pre-defined for each background mode to save computational cost and does not change with the amount of condensate. Therefore it would be the same across the model versions.*

On Page 17, Line 510 onward, the explanation between the different behavior is slightly ambiguous, especially the last sentence of the paragraph: "Additionally, in highly polluted areas, an increased number of particles may inhibit activation because more particles compete for the same water vapor". Does it want to say that in polluted areas, newly formed particles decrease supersaturation so much that it suppresses activation? If so, my understanding that in order to suppress the number of activated particles by increasing the number of particles, number concentrations have to be extremely high and would occur in very few parts of the atmosphere. Does also the modal representation affect this behavior? With a high number of NPF particles entering the smallest mode, the mode will shift to smaller sizes artificially suppressing activation.

— *This is a good point, and further analysis for the follow up paper has shown this to be due to a reduction in the size of the larger particles. Basically what happens is that the larger, non-NPF particles receive less condensate when NPF is high, due to the increase in condensation sink that the NPF particles represent. Thus given the same emissions, higher NPF may reduce the size of the larger particles which might otherwise have activated. This is not contrary to the original text, but rather a clarification of it. We have made the text clearer here, and deleted the last sentence quoted by the referee above:*

*"Additionally, in highly polluted areas, an increased number of particles may inhibit activation because more particles compete for the same water vapor".*

*This is done because we agree with the referee that this effect would likely be small compared to the effect due to the change in particle sizes mentioned above.*

**Technical comments**

Page 2, Line 49: Correct "depend" to "dependent"

— *This has been corrected.*

The model is referred to as CAM6-Nor and CAM-Nor6. On Page 9, Line 256, CAM6-Oslo is mentioned. Please correct the wrong namings.

— *These have all been corrected to CAM6-Nor.*